# HiCLIP: Contrastive Language-Image Pretraining with Hierarchy-aware Attention

**Shijie Geng**[1,2]*, **Jianbo Yuan**[2], **Yu Tian**[2], **Yuxiao Chen**[1], **Yongfeng Zhang**[1]

[1]Rutgers University    [2]ByteDance Inc.

`{sg1309, yc984, yongfeng.zhang}@rutgers.edu,`
`{jianbo.yuan, yutian.yt}@bytedance.com`

## Abstract

The success of large-scale contrastive vision-language pretraining (CLIP) has bene-fited both visual recognition and multimodal content understanding. The concise design brings CLIP the advantage in inference efficiency against other vision-language models with heavier cross-attention fusion layers, making it a popular choice for a wide spectrum of downstream tasks. However, CLIP does not explicitly capture the hierarchical nature of high-level and fine-grained semantics conveyed in images and texts, which is arguably critical to vision-language understanding and reasoning. To this end, we equip both the visual and language branches in CLIP with hierarchy-aware attentions, namely Hierarchy-aware CLIP (HiCLIP), to progressively discover semantic hierarchies layer-by-layer from both images and texts in an unsupervised manner. As a result, such hierarchical aggregation significantly improves the cross-modal alignment. To demonstrate the advantages of HiCLIP, we conduct qualitative analysis on its unsupervised hierarchy induction during inference, as well as extensive quantitative experiments on both visual recognition and vision-language downstream tasks.[1]

## 1 Introduction

In recent years, vision-language pretraining has achieved significant progress pairing with large-scale multimodal data. Contrastive vision-language pretraining (CLIP) features its generalization ability for zero-shot tasks and robustness to domain shift (Radford et al., 2021). Moreover, the spectrum of problems that CLIP can solve range from visual recognition, image-text retrieval, and vision-language reasoning tasks via providing appropriate prompt engineering (Zhou et al., 2022; Gao et al., 2021; Xu et al., 2021; Shridhar et al., 2021; Rao et al., 2022; Zhong et al., 2022). Since CLIP is built upon simple cross-modal interactions, it has superior inference efficiency over cross-attention based vision-language models (Li et al., 2021; Chen et al., 2020; Li et al., 2020; Tan & Bansal, 2019; Dou et al., 2022). Recent studies including DeCLIP (Li et al., 2022), SLIP (Mu et al., 2021), and FILIP (Yao et al., 2022) extend CLIP by either leveraging extra self-supervision training objectives or performing contrastive loss on dense token features.

As humans perceive the world in a hierarchical manner Hubel & Wiesel (1968); Fukushima & Miyake (1982); Kuzovkin et al. (2018), such hierarchical nature in vision and language contents has been explored to assist the design of various model architectures. However, contrastive vision-language learning methods like CLIP often cannot capture visual and linguistic hierarchies in an explicit way. In the example of right figure in Fig. 1 (a), the pixels first form local patches as image encoder's inputs, and are further merged into semantic *groups* denoting objects ("traffic lights", "sky"), attributes ("cloudy"), etc. Similarly, syntax hierarchies can also be observed in natural languages where the caption can be decomposed into *constituents* as shown in Fig. 1 (b). Therefore, we argue that the hierarchical nature (i.e., merging from local to global) in vision and language is critical and can be explicitly utilized for improving CLIP's capability on multimodal tasks, especially those requiring high-level understanding and reasoning.

---

*This work was conducted while interning at ByteDance.
[1]We release our implementation of HiCLIP at `https://github.com/jeykigung/HiCLIP`.

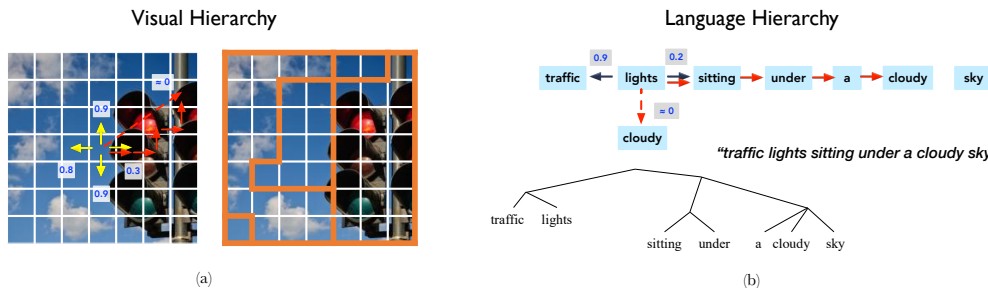

Figure 1: Illustration of hierarchical structures in both (a) vision and (b) language modalities. Based on the affinity scores between adjacent vision patches or word tokens (marked in *blue boxes*), the attention mask $C$ in hierarchy-aware attention considers both spatial and semantic similarity following the highest valued route (marked in *red arrows*) between two patches or tokens. The affinity scores evolve layer-by-layer contributing to the different levels of hierarchy granularity.

To this end, we introduce *hierarchy-aware attention* into CLIP, denoted as HiCLIP. Hierarchy-aware attention applies an attention mask to the conventional attention mechanism to indicate the tendency to merge certain vision patches and language tokens into groups because they are spatially and semantically or visually similar. We generalize hierarchy-aware attention to both images and texts, where its mask is obtained by first calculating the neighbouring affinity score among adjacent patches or tokens, and then propagating the scores across any given patch or token pairs. In addition, we formulate the affinity score with an increasing trend as the layer gets deeper to ensure the merged groups remains the same. In this way, we progressively aggregate hierarchies in a layer-by-layer manner for both images and texts.

To be specific, for modeling hierarchies in natural language, we share similar intuitions with the previous studies on unsupervised grammar induction, which aim at unsupervised hierarchical mining (Shi et al., 2019; Drozdov et al., 2019). Tree Transformer (Wang et al., 2019) proposes a similar modified attention mechanism which is essentially a special case of hierarchy-aware attention, where the attention mask is instantiated as constituent prior to encourage the merging of semantically similar tokens. Capturing hierarchies in visual contents is more challenging, because spatial correlation should also be considered in addition to visual similarities. Therefore, we extend the hierarchy-aware attention to Vision Transformers (Dosovitskiy et al., 2021) by creating a *Group Transformer* to progressively aggregate image patches into semantic groups until all patches are merged in one common group which is the original image. Different from the 1D scenario in Tree Transformer, the neighboring affinity score is computed among the four adjacent neighbors of each image patch (Fig. 1 (a)). Afterwards, we propagate neighboring affinity scores by comparing two special paths connecting image patches on the 2D grid graph.

When we apply such hierarchy-aware attention to both image and text branches in CLIP, we obtain the proposed hierarchy-aware CLIP (HiCLIP) which features the following advantages: (1) it is able to automatically discover hierarchies in vision and language that matches human intuitions in an unsupervised manner; (2) it generates better multimodal representations especially for vision-language downstream tasks; and (3) it features a comprehensive hierarchy visualization to help parse visual and textual hierarchies. To prove the aforementioned advantages, we pretrain HiCLIP and other CLIP-style approaches on large-scale image-text pairs, and conduct extensive experiments on downstream tasks including visual recognition, image-text retrieval, visual question answering, and visual entailment reasoning. To sum up, our contributions are summarized as follows:

- We incorporate hierarchy-aware attention into CLIP (HiCLIP) for both image and text contents, which achieves better performances on vision and vision-language downstream tasks.
- To model images in a hierarchical manner, we propagate neighboring affinity scores through two special paths on 2D grid graphs and generalize the hierarchy-aware attention to Vision Transformer.
- We visualize the evolution of hierarchies in images and texts to demonstrate the ability of unsupervised hierarchy induction of HiCLIP, which contributes to a better interpretability.

## 2 RELATED WORK

**Vision-Language Models.** With the proliferation of multimodal information, exploring the interaction of vision and language information becomes an important topic. As a result, many vision-language

models have flourished recently. Based on how the training objective is designed, they can be divided into three categories. The first category includes early bilinear pooling and attention based multimodal models such as MUTAN (Ben-Younes et al., 2017), BAN (Kim et al., 2018), bottom-up top-down attention (Anderson et al., 2018), and intra-inter modality attention (Gao et al., 2019). The second category is built upon the masked language modeling (MLM) pretraining objective and consists of approaches such as ViLBERT (Lu et al., 2019), LXMERT (Tan & Bansal, 2019), UNITER (Chen et al., 2020) and ALBEF (Li et al., 2021). Several recent approaches such as SOHO (Huang et al., 2021) and BEiT (Wang et al., 2022) futhur extends MLM to masked visual modeling (MVM) to push the boundary of multimodal learning. In addition, CLIP family models (Radford et al., 2021; Li et al., 2022; Mu et al., 2021; Yao et al., 2022; Chen et al., 2023) that rely on vision-language contrastive learning and large-scale image-text pairs constitutes the last category.

**Unsupervised Grammar Induction.** Unsupervised grammar induction is a classic topic in NLP domain aiming at automatically inducing phrase-structure grammars from free-text without parse tree annotations. In the earlier age, probabilistic context free grammars (PCFGs) built upon context-free grammar is widely applied and are solved by inside-outside algorithm (Baker, 1979) or CYK algorithm (Sakai, 1961). More recently, many deep learning based approaches have been proposed by extending the conventional methods into neural networks such as C-PCFG (Kim et al., 2019) and DIORA (Drozdov et al., 2019), designing special modules to induce tree structures such as PRPN (Shen et al., 2018), ON-LSTM (Shen et al., 2019), Tree Transformer (Wang et al., 2019), or assisting unsupervised grammar induction with the help of cross-modality alignment such as VG-NSL (Shi et al., 2019), VC-PCFG (Zhao & Titov, 2020), and CLIORA (Wan et al., 2022).

**Hierarchical Discovery in Vision.** Discovering the hierarchy in visual contents is a well-established area of vision research. For example, Lin et al. (2017) constructs the hierarchy of feature pyramids in object detection to help the model capture semantics at all scales. More recent work on transformers (Liu et al., 2021; Zhang et al., 2020) also adopts similar intuition to generate hierarchical feature maps with special local-global attention designs. Meanwhile, another line of research on designing new fine-grained parsing tasks aims to understand hierarchy within a scene, such as scene graph parsing (Krishna et al., 2017; Zhang et al., 2019) and action graph parsing (Ji et al., 2020). Recently, more efforts are devoted to automatic hierarchy learning with self-supervised or weakly-supervised objectives (Xie et al., 2021; Dai et al., 2021), designing special inductive bias for the self-attention mechanism (Yu et al., 2022; Zheng et al., 2021), and automatically merging semantically similar embeddings (Xu et al., 2022; Locatello et al., 2020). Our work has the scope of developing special attention constraint and utilizing contrastive learning objectives for unsupervised hierarchy discovery.

## 3 HIERARCHY-AWARE ATTENTION IN CLIP

As discussed in Section 1, both vision and language share a hierarchical nature in information parsing. The lower level of the hierarchy contains more localized and finer-grained information while the higher levels capture more holistic semantics. These properties are in line with how we humans understand vision (Hubel & Wiesel, 1968; Fukushima & Miyake, 1982; Kuzovkin et al., 2018) and language information (Chomsky, 1956; Manning, 2022).

### 3.1 A FRAMEWORK OF HIERARCHICAL INFORMATION AGGREGATION

Hierarchy-aware attention is based on the attention mechanism in conventional Transformers. Given query $Q$, key $K$, and value $V$, and the scaling factor $\sqrt{d_h}$ that maintains the order of magnitude in features where $d_h$ denotes the feature dimension, the general Attention function is defined as:

$$\text{Attention}(Q, K, V) = \text{softmax}\left(\frac{QK^\top}{\sqrt{d_h}}\right)V \tag{1}$$

As illustrated in Fig. 2, we propose to enhance CLIP's vision and language branch with a hierarchy-aware attention. Following the common transformer architecture, given the modality inputs being split into low-level image patches and text tokens, we recursively merge patches and tokens that are semantically and spatially similar, and gradually form more semantic-concentrated clusters such as image objects and text phrases. First, we define the hierarchy aggregation priors as follows:

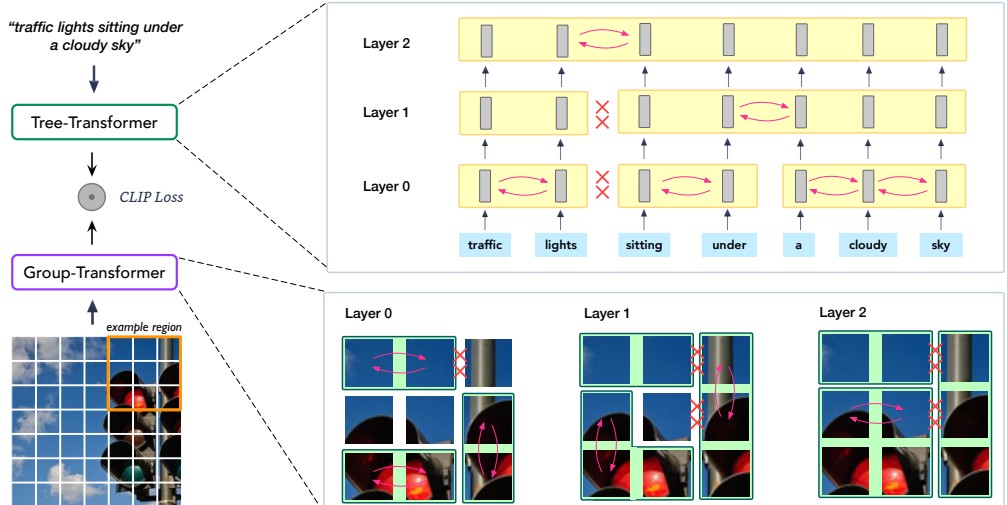

Figure 2: Illustration of Hierarchy-aware CLIP (HiCLIP), which employs hierarchy-aware attention to both vision and language encoders. HiCLIP estimates the affinity scores of neighbouring vision patches or language tokens and progressively groups them into higher-level constituents, encouraging encoders to explicitly capture hierarchical information during training.

**Tendency to merge.** We recursively merge patches and tokens into higher-level clusters that are *spatially and semantically* similar. Intuitively, if two nearby image patches have similar appearances, it is natural to merge them as one to convey the same semantic information.

**Non-splittable.** Once the patches or tokens are merged, they will *never be split* at later layers. With this constraint, we aim to enforce that the hierarchical information aggregation will never get degraded, and as a result, preserve the complete process of hierarchy evolving layer-by-layer.

We then incorporate these hierarchy aggregation priors into an attention mask $C$ which serves as an extra inductive bias to help the conventional attention mechanism in Transformers to better explore the hierarchical structures adaptive to each modality format, i.e., 2D grid on images and 1D sequence on texts. Therefore, the proposed hierarchy-aware attention can be defined as:

$$\text{Hierarchy\_Attention} = \left( C \odot \text{softmax} \left( \frac{QK^T}{\sqrt{d_h}} \right) \right) V. \tag{2}$$

Note that $C$ is shared among all heads and progressively updated bottom-up across Transformer layers. We elaborate on the formulations of the hierarchy-aware mask $C$ for each modality as follows.

### 3.1.1 HIERARCHY INDUCTION FOR LANGUAGE BRANCH

In this section, we revisit the tree-transformer method from the proposed hierarchy-aware attention point of view and show how to impose hierarchy aggregation priors on $C$ with three steps.

*Generate neighboring attention score.* Neighboring attention score describes the merging tendency of adjacent word tokens. Two learnable key, query matrices $W'_Q$, $W'_K$ are adopted to transfer any adjacent word tokens $(t_i, t_{i+1})$, so that the neighboring attention score $s_{i,i+1}$ is defined as their inner-product:

$$s_{i,i+1} = \frac{(t_i W'_Q) \cdot (t_{i+1} W'_K)}{\sigma_t}. \tag{3}$$

Here $\sigma_t$ is a hyper-parameter to control the scale of the generated scores. Then, for each token $t_i$, a $\text{softmax}$ function is employed to normalize its merging tendency of two neighbors:

$$p_{i,i+1}, p_{i,i-1} = \text{softmax} \left( s_{i,i+1}, s_{i,i-1} \right). \tag{4}$$

For neighbor pairs $(t_i, t_{i+1})$, the neighboring affinity score $\hat{a}_{i,i+1}$ is defined as the geometric mean of $p_{i,i+1}$ and $p_{i+1,i}$: $\hat{a}_{i,i+1} = \sqrt{p_{i,i+1} \cdot p_{i+1,i}}$. From a graph perspective, it describes the strength of edge $e_{i,i+1}$ by comparing it with edges $e_{i-1,i}$ ($p_{i,i+1}$v.s. $p_{i,i-1}$) and $e_{i+1,i+2}$ ($p_{i+1,i}$v.s. $p_{i+1,i+2}$).

*Enforcing Non-splittable property.* Intuitively, a higher neighboring affinity score indicates that two neighbor tokens are more closely bonded. To assure merged tokens will not be splitted, layer-wise affinity scores $a_{i,i+1}^l$ should increase as the network goes deeper, i.e., $a_{i,i+1}^l \geq a_{i,i+1}^{l-1}$ for all $l$, to help gradually generate a hierarchy structure as desired:

$$a_{i,i+1}^l = a_{i,i+1}^{l-1} + \left(1 - a_{i,i+1}^{l-1}\right) \hat{a}_{i,i+1}^l, \tag{5}$$

*Modeling the tendency to merge.* To measure the tendency to merge, namely $C_{i,j}$, for *any* word token pair $(t_i, t_j)$, we propagate the affinity scores of neighboring tokens between $(t_i, t_j)$. Specifically, $C_{i,j}$ is derived through the multiplication operation as $C_{i,j} = \prod_{k=i}^{j-1} a_{k,k+1}$. Note that $C$ is a symmetric matrix, so we have $C_{i,j} = C_{j,i}$.

### 3.1.2 HIERARCHY INDUCTION FOR VISUAL BRANCH

From a graph perspective, it is easier to generalize the hierarchy-aware mask $C$ from the 1D sequence in language to the 2D grid in vision domain. First, we also employ query and key matrices $W_Q''$, $W_K''$ to calculate the neighboring attention scores among the four-adjacency neighbors of each patch $t_{i,j}$:

$$s_{(i,j),(i',j')} = \frac{(t_{i,j} W_Q'') \cdot (t_{i',j'} W_K'')}{\sigma_v}, \tag{6}$$

where $\sigma_v$ is used to control the scale of the generated scores, and the neighborhood $t_{i',j'}$ is limited to the four-adjacency patches of $t_{i,j}$ as $(i', j') \in \{(i + \delta, j + \eta); \delta, \eta \in \{-1, +1\}\} \equiv \mathcal{A}$.

Next, for each patch $p_{i,j}$, the softmax normalizing function is employed to get the merging tendency of $t_{i,j}$ to its four neighbors:

$$\{p_{(i,j),(i',j')}\} = \text{softmax}(\{s_{(i,j),(i',j')}; (i', j') \in \mathcal{A}\}). \tag{7}$$

Similar to the formulation in the language branch, the neighboring affinity score with non-splittable property can be obtained by:

$$a_{(i,j),(i',j')}^l = a_{(i,j),(i',j')}^{l-1} + \left(1 - a_{(i,j),(i',j')}^{l-1}\right) \hat{a}_{(i,j),(i',j')}^l, \tag{8}$$

where $\hat{a}_{(i,j),(i',j')} = \sqrt{p_{(i,j),(i',j')} \cdot p_{(i',j'),(i,j)}}$.

Lastly, the neighboring affinity scores needs to be propagated to the whole image to acquire $C_{(i_1,j_1),(i_2,j_2)}$ between *any* two patches $(t_{i_1,j_1}, t_{i_2,j_2})$. As the image can be seen as a 2D-grid graph, a natural solution is considering $C_{(i_1,j_1),(i_2,j_2)}$ as the length of the shortest path with $-\log(a_{(i,j),(i',j')})$ as the edge weights. To achieve better computational efficiency, we consider two special paths: connecting $(t_{i_1,j_1}, t_{i_2,j_2})$ along the grid with only one turn. The length of these two paths can be calculated by horizontal and vertical propagation as follows:

$$C_1, C_2 = \prod_{n=i_1}^{i_2-1} a_{(n,j_1),(n+1,j_1)} \prod_{m=j_1}^{j_2-1} a_{(i_2,m),(i_2,m+1)}, \prod_{m=j_1}^{j_2-1} a_{(i_1,m),(i_1,m+1)} \prod_{n=i_1}^{i_2-1} a_{(n,j_2),(n+1,j_2)}, \tag{9}$$

and $C_{(i_1,j_1),(i_2,j_2)} = \max(C_1, C_2)$. Intuitively, $C_{(i_1,j_1),(i_2,j_2)}$ finds the maximum merging tendency along two possible paths, either horizontal-first or vertical-first. In this way, both spatial and visual similarities have contributed to the attention mask $C$ for 2D images. Since our approach tries to organize vision patches sharing high similarities into groups, we thus dub it "Group Transformer".

*Relation to Recursive Bilateral Filtering.* Recursive bilateral filtering (Yang, 2012) shares similar spirit with our Group Transformer. Given two pixels in an image, recursive bilateral filtering decomposes the calculation of range filtering kernel $R$ into two 1D operations – horizontal and vertical. For each 1D operation, let $x_k, x_i$ denote two pixels on a scanline of the 2D image, the 1D range filtering kernel $R_{k,i} = R_{k,k+1} R_{k+1,k+2} \cdots R_{i-2,i-1} R_{i-1,i} = \prod_{j=k}^{i-1} R_{j,j+1}$, where $R_{j,j+1}$ is computed with a radial basis function kernel: $R_{j,j+1} = \exp\left(-\frac{|x_j - x_{j+1}|^2}{2\sigma_R^2}\right)$. We can observe the similar property that Tree Transformer possesses. As bilateral filters tend to preserve sharp edges while smoothing other parts of an image, this is in accordance with the goal of our Group Transformer – aggregating similar patches into a group. The differences between recursive bilateral filtering and our framework are mainly two perspectives: 1) the basic operation unit is pixel in recursive bilateral filtering while our approach uses either patch or word token; 2) radial basis function kernel is employed to measure the range similarity in recursive bilateral filtering while we use neighboring affinity score instead.

## 3.2 HIERARCHY-AWARE CLIP

*Pretraining with HiCLIP.* To equip both CLIP branches with the ability of dynamic hierarchy discovery, our Hierarchy-aware CLIP adopts Group Transformer as the image encoder and employs Tree Transformer as the text encoder. Let $\mathbf{v}$ and $\mathbf{u}$ denote the image and text feature vectors, the contrastive pretraining objective $\mathscr{L}$ can be written as:

$$\mathscr{L} = -\frac{1}{N}\sum_i^N \log \frac{\exp\left(\mathbf{v}_i^\top \mathbf{u}_i/\tau\right)}{\sum_{j=1}^N \exp\left(\mathbf{v}_i^\top \mathbf{u}_j/\tau\right)} - \frac{1}{N}\sum_i^N \log \frac{\exp\left(\mathbf{u}_i^\top \mathbf{v}_i/\tau\right)}{\sum_{j=1}^N \exp\left(\mathbf{u}_i^\top \mathbf{v}_j/\tau\right)} \qquad (10)$$

where $\tau$ is a learnable temperature parameter, and $N$ is the total number of image-text pairs.

*Unsupervised Hierarchy Induction.* During inference, we follow Eq. (5) and Eq. (8) to generate all neighboring affinity scores $\{a_{i,i+1}^l\}_{l=1}^L$ and $\{a_{(i,j),(i',j')}^l\}_{l=1}^L$ from bottom to top layer $L$ for the texts and images, respectively. These neighboring affinity scores are then used for hierarchy induction. Intuitively, a low affinity score at a certain layer indicates the two corresponding neighbours remain split within this layer. When repeating such process in a top-down greedy manner, we are able to generate tree hierarchies for texts and similar group hierarchies for images in an unsupervised fashion.

## 4 EXPERIMENTS

### 4.1 EXPERIMENTAL SETTINGS

**Pretraining Datasets** To make a fair comparison with the state-of-the-art contrastive vision-language pretraining approaches, we adopt the **YFCC15M** benchmark proposed in (Cui et al., 2022) which builds on a subset from YFCC100M (Thomee et al., 2016) consisting of 15M image-text pairs. In addition, we construct a **30M** version of pretraining data by including Conceptual Caption 3M (CC3M) (Sharma et al., 2018) and 12M (CC12M) (Changpinyo et al., 2021). We thus validate our model on the two different scales of pretraining data.

**Downstream Datasets** Following CLIP and DeCLIP, we select 11 visual recognition datasets under the zero-shot setting, namely ImageNet (Deng et al., 2009), CIFAR 10 & CIFAR 100 (Krizhevsky et al., 2009), StanfordCars (Krause et al., 2013), Caltech101 (Fei-Fei et al., 2004), Flowers102 (Nilsback & Zisserman, 2008), SUN397 (Xiao et al., 2010), DTD (Cimpoi et al., 2014), FGVCAircraft (Maji et al., 2013), OxfordPets (Parkhi et al., 2012), and Food101 (Bossard et al., 2014). Same zero-shot classification protocol is applied following Radford et al. (2021) which uses predefined prompts as text inputs. The full list of used prompts is provided in the Appendix. Although CLIP and DeCLIP only evaluates on visual recognition, we also provide comprehensive comparisons on vision-language tasks which are more desired in evaluating multimodal models, including: image-text retrieval on MSCOCO Caption (Chen et al., 2015), as well as vision-language reasoning on VQAv2 (Antol et al., 2015) and SNLI-VE (Xie et al., 2019).

**Implementation Details** Two variants of Vision Transformer (Dosovitskiy et al., 2021) are used as the image encoder in our experiments – ViT-B/32 and ViT-B/16, while the text encoder is a vanilla Transformer (Vaswani et al., 2017) following CLIP as a fair comparison. The embedding size of both image and text features are 512 throughout our paper. To make a fair comparison with CLIP family baselines, we train all models for 32 epochs under the same set of pretraining hyperparameters including learning rate, warmup steps, weight decay, etc. The input image size is set to $224 \times 224$, and the input text sequence length is truncated or padded to 77. The scaling factor $\sigma_t$ and $\sigma_v$ of Hierarchy-aware attention are both set to 256 for Group Transformer and Tree Transformer. Following CLIP and DeCLIP, the learnable temperature parameter $\tau$ is initialized as 0.07.

### 4.2 VISUAL RECOGNITION

We first compare HiCLIP with state-of-the-art CLIP family approaches on YFCC15M benchmark (Cui et al., 2022) containing only ImageNet zero-shot setting. Then we present zero-shot classification results on the other common visual recognition datasets. Both results are presented in Table 1.

*YFCC15M Benchmark.* CLIP Radford et al. (2021), SLIP Mu et al. (2021), and FILIP Yao et al. (2022) all leverage a contrastive learning and can be directly compared with HiCLIP. When limiting

Table 1: Zero-shot classification top-1 accuracy (%) on 11 vision datasets against state-of-the-art CLIP-style models, including: CIFAR10/100 (C10/100), Food101 (F101), Flowers (Flow), Caltech (Cal), Aircraft (Air), and ImageNet (IN). ViT-B/32 is used for all compared models.

| Model | Data | C10 | C100 | F101 | Pets | Flow. | SUN | Cars | DTD | Cal. | Air. | IN | Avg. |
|---|---|---|---|---|---|---|---|---|---|---|---|---|---|
| CLIP | 15M | 63.7 | 33.2 | 34.6 | 20.1 | 50.1 | 35.7 | 2.6 | 15.5 | 59.9 | 1.2 | 32.8 | 31.8 |
| SLIP | 15M | 50.7 | 25.5 | 33.3 | 23.5 | 49.0 | 34.7 | 2.8 | 14.4 | 59.9 | 1.7 | 34.3 | 30.0 |
| FILIP | 15M | 65.5 | 33.5 | 43.1 | 24.1 | 52.7 | 50.7 | 3.3 | 24.3 | 68.8 | 3.2 | 39.5 | 37.2 |
| HiCLIP | 15M | 74.1 | 46.0 | 51.2 | 37.8 | 60.9 | 50.6 | 4.5 | 23.1 | 67.4 | 3.6 | **40.5** | **41.8 (↑ +10.0)** |
| DeCLIP | 15M | 66.7 | 38.7 | 52.5 | 33.8 | 60.8 | 50.3 | 3.8 | 27.7 | 74.7 | 2.1 | 43.2 | 41.3 |
| DeFILIP | 15M | 70.1 | 46.8 | 54.5 | 40.3 | 63.7 | 52.4 | 4.6 | 30.2 | 75.0 | 3.3 | 45.0 | 44.2 |
| HiDeCLIP | 15M | 65.1 | 39.4 | 56.3 | 43.6 | 64.1 | 55.4 | 5.4 | 34.0 | 77.0 | 4.6 | **45.9** | **44.6 (↑ +3.3)** |
| CLIP | 30M | 77.3 | 48.1 | 59.1 | 58.5 | 58.2 | 52.6 | 17.7 | 28.0 | 80.8 | 3.2 | 48.8 | 48.4 |
| HiCLIP | 30M | 77.6 | 56.2 | 63.9 | 65.6 | 62.5 | 60.7 | 22.2 | 38.0 | 82.4 | 5.5 | **52.9** | **53.4 (↑ +5.0)** |
| DeCLIP | 30M | 84.0 | 57.1 | 67.3 | 71.7 | 65.0 | 62.5 | 23.0 | 39.5 | 86.1 | 5.3 | 55.3 | 56.1 |
| HiDeCLIP | 30M | 80.4 | 54.2 | 68.9 | 73.5 | 66.1 | 65.2 | 26.8 | 44.2 | 87.8 | 7.2 | **56.9** | **57.4 (↑ +1.3)** |

the comparisons within this scope, HiCLIP achieves the largest performance gain (7.7%) over the other CLIP-style models. Since DeCLIP Li et al. (2022) and DeFILIP Cui et al. (2022) apply multiple single-modal self-supervised tasks in addition to CLIP, we incorporated the same objectives into our hierarchy-aware model for a fair comparison (denoted as HiDeCLIP). By combining the contrastive learning and self-supervised learning loss functions, our HiDeCLIP further improves the zero-shot ImageNet classification performance by 2.7% over DeCLIP, and overall 13.1% higher than CLIP.

*11 Visual Recognition Benchmarks.* Note that we included both versions of training data, YFCC15M (short as 15M) and 30M, in this experiments as discussed in Section 4.1. We observed that the zero-shot performance on Cars and Aircraft datasets are very low for all models, because in the YFCC benchmark there are 0.04% and 0% of descriptions contains aircraft and car labels used in these datasets, such as *"Audi V8 Sedan 1994"*. HiCLIP achieves significant improvement in average over CLIP on both pretraining datasets, indicating that HiCLIP maintains substantial advantage over CLIP when scaling up the size of training data. Despite the fact that the absolute improvements by incorporating hierarchy-aware attentions into CLIP is relatively less significant than adding multiple self-supervised tasks, it is interesting that hierarchy-aware attention is compatible with self-supervised learning (DeHiCLIP) and further achieves performance improvement over DeCLIP.

## 4.3 PERFORMANCE COMPARISON ON VISION-LANGUAGE TASKS

In Table 2, we compare different CLIP-style methods on downstream vision-language tasks, including image-text retrieval which emphasizes on cross-modal alignment and two vision-language reasoning tasks (VQA and SNLI-VE) which focus more on collaborative multimodal reasoning.

*Zero-shot Image-Text Retrieval on MSCOCO.* On all different algorithms and training datasets, HiCLIP and HiDeCLIP improve the retrieval performance by a large margin. It is worth noting that without complicated self-supervised learning objectives, HiCLIP constantly outperforms DeCLIP when merely relying on CLIP's contrastive loss which is different from visual recognition tasks. This finding suggests that the benefits brought by adding self-supervised learning is effective within the scope of visual recognition, while our approach fully explores the hierarchical nature of multimodal contents which contributes to a significantly performance boost in vision-language tasks.

*Fine-tuning on Vision-Language Reasoning Tasks.* Similar to the results on zero-shot retrieval, we observe consistent performance gains for all visual reasoning tasks and pretraining data, indicating that hierarchy-aware attentions are more efficient multimodal learners and is capable of tackling tasks that require content understanding and reasoning capabilities.

## 4.4 ABLATION STUDY

In this section, we provide additional ablation studies on influence factors including the patch granularity of visual encoder and the training data volumes in Table 3. We report the following experimental results including zero-shot accuracy on ImageNet and averaged accuracy on all 11

Table 2: Zero-shot image-text retrieval on MSCOCO (5K) dataset and vision-language reasoning on VQAv2 and SNLI-VE with fine-tuning. ViT-B/32 is adopted for all models.

| Method | Data | Text Retrieval | | | Image Retrieval | | | RSum | VQA (test-dev) | | | | SNLI (val+test) |
|---|---|---|---|---|---|---|---|---|---|---|---|---|---|
| | | R@1 | R@5 | R@10 | R@1 | R@5 | R@10 | | Y/N | Num. | Other | All | Acc. |
| CLIP | 15M | 21.4 | 44.7 | 56.4 | 13.7 | 32.4 | 42.9 | 211.5 | 67.3 | 30.5 | 32.7 | 46.7 | 62.5 |
| HiCLIP | 15M | 34.2 | 60.3 | 70.9 | 20.6 | 43.8 | 55.3 | **285.1** | 69.4 | 33.7 | 37.2 | **50.1** | **67.7** |
| DeCLIP | 15M | 29.1 | 55.2 | 66.6 | 19.0 | 41.2 | 53.1 | 264.2 | 70.3 | 34.9 | 36.9 | 50.4 | 66.1 |
| HiDeCLIP | 15M | 38.7 | 64.4 | 74.8 | 23.9 | 48.2 | 60.1 | **310.1** | 72.4 | 36.1 | 40.9 | **53.3** | **70.5** |
| CLIP | 30M | 34.8 | 63.3 | 73.9 | 23.3 | 46.9 | 58.6 | 300.8 | 69.7 | 34.8 | 37.8 | 50.6 | 66.9 |
| HiCLIP | 30M | 43.9 | 69.1 | 78.8 | 27.0 | 51.8 | 62.9 | **333.5** | 72.2 | 36.1 | 40.9 | **53.2** | **70.1** |
| DeCLIP | 30M | 41.3 | 68.8 | 79.3 | 25.6 | 50.7 | 62.3 | 328.0 | 71.3 | 35.4 | 39.7 | 52.2 | 69.0 |
| HiDeCLIP | 30M | 48.6 | 74.1 | 82.7 | 29.6 | 54.9 | 66.3 | **356.2** | 73.3 | 37.0 | 42.5 | **54.6** | **72.5** |

Table 3: Ablations on the patch granularity and pretraining data scale for HiCLIP & HiDeCLIP. Rsum is the summation of the R@1, R@5, R@10 of image-to-text and text-to-image retrievals.

| Method | Encoder | Data | ImageNet Acc. | 11 Datasets Avg. | COCO Rsum | VQA Acc. | SNLI Acc. |
|---|---|---|---|---|---|---|---|
| CLIP | ViT-B/32 | 15M | 32.8 | 31.8 | 211.5 | 46.7 | 62.5 |
| HiCLIP | ViT-B/32 | 15M | 40.5 | 41.8 | 285.1 | 50.1 | 67.7 |
| DeCLIP | ViT-B/32 | 15M | 43.2 | 41.3 | 264.2 | 50.4 | 66.1 |
| HiDeCLIP | ViT-B/32 | 15M | 45.9 | 44.6 | 310.1 | 53.3 | 70.5 |
| CLIP | ViT-B/16 | 15M | 39.3 | 35.5 | 245.0 | 48.8 | 63.8 |
| HiCLIP | ViT-B/16 | 15M | 45.2 | 44.9 | 313.9 | 51.2 | 69.0 |
| DeCLIP | ViT-B/16 | 15M | 48.2 | 43.7 | 290.3 | 51.5 | 67.3 |
| HiDeCLIP | ViT-B/16 | 15M | 51.1 | 48.3 | 339.6 | 54.4 | 71.3 |
| CLIP | ViT-B/32 | 30M | 48.8 | 48.4 | 300.8 | 50.6 | 66.9 |
| HiCLIP | ViT-B/32 | 30M | 52.9 | 53.4 | 333.5 | 53.2 | 70.1 |
| DeCLIP | ViT-B/32 | 30M | 55.3 | 56.1 | 328.0 | 52.2 | 69.0 |
| HiDeCLIP | ViT-B/32 | 30M | 56.9 | 57.4 | 356.2 | 54.6 | 72.5 |

visual recognition dataset, Rsum over recall@1, 5, 10 on zero-shot image-text retrieval, as well as accuracy on VQA and SNLI with fine-tuning. In addition, we conduct component analysis in Table 4 to show that Group Transformer and Tree Transformer both play important roles in HiCLIP.

**On Patch Granularity.** We compare all downstream tasks using ViT-B/32 and ViT-B/16 as visual encoders. Since the Group Transformer is based on visual patches and benefits from finer-grained patch segments, we expect HiCLIP and HiDeCLIP achieves consistent performance improvements when directly comparing the same method across different visual encoder variants. When we fix the visual encoder and compare HiCLIP and HiDeCLIP with their corresponding baselines (i.e., CLIP and HiCLIP), HiCLIP and HiDeCLIP constantly outperform CLIP and DeCLIP on all tasks with the help of hierarchical-aware attention. It is worth noting that HiCLIP alone without complex self-supervised losses outperforms DeCLIP on three out of five tasks, with the exceptions on ImageNet and VQA by a small margin. This shows that hierarchical information captured by HiCLIP potentially benefits more to the vision-language contrastive learning paradigm.

**On Pretraining Data Scale.** As shown in Table 3, for most vision recognition tasks, we observe that the benefits contributed by a better modeling strategy saturates when more data is used during pretraining, which is in line with the findings reported by many other works including CLIP Radford et al. (2021). One possible explanation is that, in order to achieve further improvements on visual recognition tasks, a more vision-specific training scheme such as self-supervised learning potentially benefits more, because the ability of multimodal high-level reasoning are not as critical in vision-only tasks. In contrast, by scaling up the pretraining data, the performance improvements achieved on vision-language tasks are more significant and consistent across all methods. Similarly, HiCLIP and HiDeCLIP still enjoys large improvements against CLIP and DeCLIP when the pretraining dataset scales up. In addition, HiCLIP pretrained on 30M data achieves better vision-language performances on all three tasks over DeCLIP suggesting a potential better scalability of HiCLIP on vision-language reasoning tasks, while DeCLIP features better vision recognition performances.

**On Component Analysis.** In Table 4, we demonstrate that using Group Transformer alone (HiCLIP-Group) for vision modeling yields comparable improvements on visual recognition task (zero-shot ImageNet classification) with using Tree Transformer alone (HiCLIP-Tree). In addition, the

Table 4: Ablations on the use of Group Transformer (G-Trans) and Tree Transformer (T-Trans) in HiCLIP. All models are pretrained on YFCC15M.

| Method | Encoder | G-Trans | T-Trans | ImageNet Acc. | 11 Datasets Avg. | Text Retrieval | | | Image Retrieval | | | COCO Rsum |
|---|---|---|---|---|---|---|---|---|---|---|---|---|
| | | | | | | R@1 | R@5 | R@10 | R@1 | R@5 | R@10 | |
| CLIP | ViT-B/32 | - | - | 32.8 | 31.8 | 21.4 | 44.7 | 56.4 | 13.7 | 32.4 | 42.9 | 211.5 |
| HiCLIP | ViT-B/32 | - | ✓ | 37.1 | 38.4 | 28.7 | 53.8 | 65.7 | 17.2 | 38.5 | 50.3 | 254.2 |
| HiCLIP | ViT-B/32 | ✓ | - | 36.2 | 35.3 | 22.9 | 47.5 | 59.4 | 14.8 | 34.2 | 45.1 | 223.9 |
| HiCLIP | ViT-B/32 | ✓ | ✓ | **40.5** | **41.8** | 34.2 | 60.3 | 70.9 | 20.6 | 43.8 | 55.3 | **285.1** |
| CLIP | ViT-B/16 | - | - | 39.3 | 35.5 | 26.1 | 52.0 | 64.6 | 16.5 | 37.3 | 48.5 | 245.0 |
| HiCLIP | ViT-B/16 | - | ✓ | 40.4 | 39.6 | 32.8 | 58.5 | 69.7 | 19.6 | 42.3 | 54.0 | 276.9 |
| HiCLIP | ViT-B/16 | ✓ | - | 42.2 | 37.7 | 28.5 | 53.2 | 65.2 | 18.1 | 39.8 | 51.3 | 256.1 |
| HiCLIP | ViT-B/16 | ✓ | ✓ | **45.2** | **44.9** | 39.0 | 65.7 | 76.4 | 24.0 | 48.7 | 60.1 | **313.9** |

improvements on image-text retrieval are more significant when applying Tree Transformer alone than applying Group Transformer, indicating that language modeling may have more potential impact than visual modeling with regard to such vision-language tasks. Moreover, when we activate both Group Transformer and Tree Transformer, substantial performance boosts are obtained against HiCLIP-Group and HiCLIP-Tree, showcasing the synergy between dual hierarchy-aware attentions even under naive cross-modal interactions.

## 4.5 UNSUPERVISED HIERARCHY INDUCTION WITH PRETRAINED HiCLIP MODEL

It is natural to adopt Tree Transformer because texts are essentially discrete tokens among which certain semantic dependencies are shared. Following the same analogy, since each image is pre-patchified in Vision Transformers, we expect the image patches to join semantic groups gradually from bottom layers to top layers for a better representation, although it seems to be more challenging than the language counterpart. Therefore, in addition to the performance gains achieved over various downstream tasks, we also visualize the hierarchies captured in our Group and Tree Transformers. As shown in Figure 3, by virtue of explicitly modeling the inputs with hierarchy-aware attentions during pretraining, our model is able to gradually group semantically similar neighbors, showing the ability of performing hierarchical visual and language inductions in an unsupervised manner.

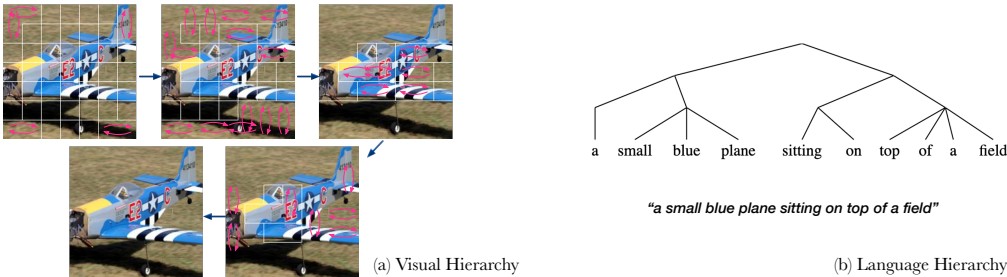

(a) Visual Hierarchy                    (b) Language Hierarchy

Figure 3: An example of unsupervised hierarchy induction for a semantically aligned image-text pair.

## 5 CONCLUSION AND FUTURE WORK

In this work, we equip both the visual and language branches of CLIP with hierarchy-aware attention to automatically capture the hierarchies from image-caption pairs. Following the discovered hierarchical structures, the proposed HiCLIP creates compact image and text embeddings via gradually aggregating spatially and semantically similar patches or tokens into common groups. Supported by extensive experiments on multiple downstream tasks, we show that hierarchy-aware attention greatly improves the alignment of image and text modalities against several recent CLIP-style approaches. Moreover, after pretraining, both branches of HiCLIP can be adopted for unsupervised hierarchy induction by analyzing the generated constituent attention weights. With limited computational resources, we conduct experiments up to 30 million image-text pairs without extensive parameter tuning. As a future direction, we plan to scale up the pretraining dataset as well as the scale of the visual encoder to fully validate the scalability of our approach. In addition, we also plan to explore the full potential of hierarchy-aware attentions with better multimodal information fusion operations compared with the simple dot product used in CLIP-style models.

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

## A  PROMPTS ENGINEERING FOR ZERO-SHOT VISUAL RECOGNITION

In this work, we follow the 80 prompts proposed in Radford et al. (2019) to evaluate zero-shot image classification on ImageNet dataset. The full list of prompts for ImageNet are presented in Table 5. For other downstream visual recognition datasets, we also use domain-specific prompts according to Radford et al. (2019). The whole prompts for the 10 downstream datasets can be found in Table 6.

## B  LINEAR PROBE PERFORMANCE

In addition to the zero-shot image classification tasks presented in Table 1, we also perform linear probe on frozen image features to estimate the quality of pretrained image encoders. We follow the settings of CLIP and DeCLIP to train a linear classifier with the L-BFGS optimizer from *scikit-learn* machine learning library. The linear probe results on downstream visual recognition tasks are presented in Table 7. From the table, we can observe that HiCLIP / HiDeCLIP still outperform CLIP / DeCLIP across all varied visual encoder types and pretraining data sizes.

## C  ADDITIONAL PRETRAINING IMPLEMENTATION DETAILS

Our implementation is based on the open-source PyTorch implementation[2]. Following Cui et al. (2022), we use AdamW optimizer (Loshchilov & Hutter, 2019) with a weight decay rate of 0.1 during pretraining. The learning rate is first linearly increased to 0.001 within 2500 warmup steps, and then decayed to 0 following the cosine strategy. For the 15M version pretraining data, we set the batch size to 4096 and run all experiments on 32 A100 GPUs. For 30M version pretraining data, we set the batch size to 8192 and run all experiments on 64 A100 GPUs.

---

[2]https://github.com/Sense-GVT/DeCLIP

Table 5: Full list of prompts to evaluate on ImageNet dataset.

| | | | |
|---|---|---|---|
| a bad photo of a {label}. | a photo of many {label}. | a sculpture of a {label}. | a photo of the hard to see {label}. |
| a low resolution photo of the {label}. | a rendering of a {label}. | graffiti of a {label}. | a bad photo of the {label}. |
| a cropped photo of the {label}. | a tattoo of a {label}. | the embroidered {label}. | a photo of a hard to see {label}. |
| a bright photo of a {label}. | a photo of a clean {label}. | a photo of a dirty {label}. | a dark photo of the {label}. |
| a drawing of a {label}. | a photo of my {label}. | the plastic {label}. | a photo of the cool {label}. |
| a close-up photo of a {label}. | a black and white photo of the {label}. | a painting of the {label}. | a painting of a {label}. |
| a pixelated photo of the {label}. | a sculpture of the {label}. | a bright photo of the {label}. | a cropped photo of a {label}. |
| a plastic {label}. | a photo of the dirty {label}. | a jpeg corrupted photo of a {label}. | a blurry photo of the {label}. |
| a photo of the {label}. | a good photo of the {label}. | a rendering of the {label}. | a {label} in a video game. |
| a photo of one {label}. | a doodle of a {label}. | a close-up photo of the {label}. | a photo of a {label}. |
| the origami {label}. | the {label} in a video game. | a sketch of a {label}. | a doodle of the {label}. |
| a origami {label}. | a low resolution photo of a {label}. | the toy {label}. | a rendition of the {label}. |
| a photo of the clean {label}. | a photo of a large {label}. | a rendition of a {label}. | a photo of a nice {label}. |
| a photo of a weird {label}. | a blurry photo of a {label}. | a cartoon {label}. | art of a {label}. |
| a sketch of the {label}. | a embroidered {label}. | a pixelated photo of a {label}. | itap of the {label}. |
| a jpeg corrupted photo of the {label}. | a good photo of a {label}. | a plushie {label}. | a photo of the nice {label}. |
| a photo of the small {label}. | a photo of the weird {label}. | the cartoon {label}. | art of the {label}. |
| a drawing of the {label}. | a photo of the large {label}. | a black and white photo of a {label}. | the plushie {label}. |
| a dark photo of a {label}. | itap of a {label}. | graffiti of the {label}. | a toy {label}. |
| itap of my {label}. | a photo of a cool {label}. | a photo of a small {label}. | a tattoo of the {label}. |

## D  MORE VISUALIZATION RESULTS & VISUALIZATION PROCESS

Besides the visualization results illustrated in Figure 3, we also provide eight more cases on unsupervised hierarchy induction from Figure 4 to Figure 11.

Moreover, we provide the detailed descriptions of the visualization process for input images in Algorithm 1, where we set the list of "break" threshold values $\{\theta_1, \ldots, \theta_{12}\}$ to $\{0.35, 0.5, 0.5, 0.6, 0.8, 0.85, 0.9, 0.9, 0.9, 0.9, 0.9, 0.9\}$. For unsupervised grammar induction, we adopt the same parsing algorithm as in Tree Transformer (Wang et al., 2019). Based on the visualization results, we can conclude that by integrating hierarchy-aware attention into the conventional attention mechanism, our HiCLIP can discover and aggregate spatially and semantically similar visual patches and language tokens in a layer-by-layer manner.

However, current unsupervised hierarchy induction of HiCLIP (visualization of vision encoder especially) follows a top-down style and relies on the threshold values to decide whether to split two adjacent visual patches and language tokens. For the visual hierarchy, we trivially specify thresholds for different layers (the higher layer also has a higher threshold value). Thus, the threshold list may not be suitable for every image. In addition, changing the threshold values may influence the visual and language induction results. It would be better if the thresholds are adaptive to every input image and sentence. Our future work is to find a better way (e.g., a data-dependent algorithm) to parse the $C$ matrix for each layer.

## E  VISUALIZATION OF LEARNED FEATURE SPACE

In Figure 12 and Figure 13, we provide the t-SNE visualization of the learned feature space for CLIP, HiCLIP and DeCLIP pretrained on *YFCC-15M* and *30M* data, respectively. We use the 10 classes of CIFAR-10 dataset to conduct all the visualization experiments.

Table 6: Full list of prompts to evaluate on 10 downstream domain-specific visual recognition datasets.

**CIFAR 10 & CIFAR 100**

| | | | |
|---|---|---|---|
| a photo of a {label}. | a blurry photo of a {label}. | a black and white photo of a {label}. | a low contrast photo of a {label}. |
| a high contrast photo of a {label}. | a bad photo of a {label}. | a good photo of a {label}. | a photo of a small {label}. |
| a photo of a big {label}. | a photo of the {label}. | a blurry photo of the {label}. | a black and white photo of the {label}. |
| a low contrast photo of the {label}. | a high contrast photo of the {label}. | a bad photo of the {label}. | a good photo of the {label}. |
| a photo of the small {label}. | a photo of the big {label}. | | |

**Food 101**

a photo of {label}, a type of food.

**Caltech101**

| | | | |
|---|---|---|---|
| a photo of a {label}. | a painting of a {label}. | a plastic {label}. | a sculpture of a {label}. |
| a sketch of a {label}. | a tattoo of a {label}. | a toy {label}. | a rendition of a {label}. |
| a embroidered {label}. | a cartoon {label}. | a {label} in a video game. | a plushie {label}. |
| a origami {label}. | art of a {label}. | graffiti of a {label}. | a drawing of a {label}. |
| a doodle of a {label}. | a photo of the {label}. | a painting of the {label}. | the plastic {label}. |
| a sculpture of the {label}. | a sketch of the {label}. | a tattoo of the {label}. | the toy {label}. |
| a rendition of the {label}. | the embroidered {label}. | the cartoon {label}. | the {label} in a video game. |
| the plushie {label}. | the origami {label}. | art of the {label}. | graffiti of the {label}. |
| a drawing of the {label}. | a doodle of the {label}. | | |

**StanfordCars**

| | | | |
|---|---|---|---|
| a photo of a {label}. | a photo of the {label}. | a photo of my {label}. | i love my {label}! |
| a photo of my dirty {label}. | a photo of my clean {label}. | a photo of my new {label}. | a photo of my old {label}. |

**DTD**

| | | | |
|---|---|---|---|
| a photo of a {label} texture. | a photo of a {label} pattern. | a photo of a {label} thing. | a photo of a {label} object. |
| a photo of the {label} texture. | a photo of the {label} pattern. | a photo of the {label} thing. | a photo of the {label} object. |

**FGVCAir-craft**

| | |
|---|---|
| a photo of a {label}, a type of aircraft. | a photo of the {label}, a type of aircraft. |

**Flowers102**

a photo of a {label}, a type of flower.

**OxfordPets**

a photo of a {label}, a type of pet.

**SUN39**

| | |
|---|---|
| a photo of a {label}. | a photo of the {label}. |

# F    DETAILED ILLUSTRATION OF THE COMPUTATION OF $C$

In Figure 14, we illustrate the detailed computation steps of the attention mask $C$. For the toy example sentence "a blue cat sitting on bench", we show how the $C_{i,j}^l$ matrix in each Tree-Transformer layer is calculated from neighbourhood affinity scores $a_{i,i+1}^l$ through the multiplication operation (i.e., $C_{i,j}^l = \prod_{k=i}^{j-1} a_{k,k+1}^l$), where $i \in \{0, \ldots, N-1\}$, $N$ is the input sequence length.

Table 7: Linear probe performance on downstream datasets. C10/100 is CIFAR10/100, F101 is Food101, Flow is Flowers, Cal is Caltech, and Air is Aircraft.

| Model | Data | C10 | C100 | F101 | Pets | Flow. | SUN | Cars | DTD | Cal. | Air. | Avg. |
|---|---|---|---|---|---|---|---|---|---|---|---|---|
| CLIP-ViT-B/32 | 15M | 86.5 | 64.7 | 69.2 | 64.6 | 90.6 | 66.0 | 24.9 | 61.3 | 79.1 | 23.1 | 63.0 |
| HiCLIP-ViT-B/32 | 15M | 89.5 | 71.1 | 73.5 | 70.6 | 91.9 | 68.8 | 30.8 | 63.9 | 84.8 | 27.4 | **67.2 (↑ +4.2)** |
| DeCLIP-ViT-B/32 | 15M | 89.2 | 69.0 | 75.4 | 72.2 | 94.4 | 71.6 | 31.0 | 68.8 | 87.9 | 27.6 | 68.7 |
| HiDeCLIP-ViT-B/32 | 15M | 88.1 | 70.7 | 77.6 | 75.5 | 95.6 | 72.2 | 36.0 | 70.1 | 90.0 | 32.6 | **70.8 (↑ +2.1)** |
| CLIP-ViT-B/16 | 15M | 88.5 | 66.4 | 77.2 | 69.3 | 94.1 | 69.8 | 29.0 | 65.2 | 82.4 | 25.5 | 66.7 |
| HiCLIP-ViT-B/16 | 15M | 89.1 | 70.4 | 81.0 | 75.3 | 95.2 | 72.5 | 36.4 | 68.7 | 86.4 | 32.3 | **70.7 (↑ +4.0)** |
| DeCLIP-ViT-B/16 | 15M | 88.7 | 69.5 | 83.0 | 74.3 | 97.3 | 74.4 | 36.9 | 70.9 | 89.8 | 32.2 | 71.7 |
| HiDeCLIP-ViT-B/16 | 15M | 88.8 | 70.3 | 84.3 | 80.6 | 97.1 | 75.1 | 42.5 | 74.3 | 90.7 | 38.3 | **74.2 (↑ +2.5)** |
| CLIP-ViT-B/32 | 30M | 92.0 | 74.7 | 78.8 | 80.7 | 93.7 | 72.6 | 55.9 | 71.4 | 88.6 | 29.7 | 73.8 |
| HiCLIP-ViT-B/32 | 30M | 92.8 | 75.8 | 80.5 | 81.3 | 94.4 | 73.6 | 59.4 | 72.2 | 90.3 | 33.6 | **75.4 (↑ +1.6)** |
| DeCLIP-ViT-B/32 | 30M | 93.1 | 76.9 | 82.0 | 82.7 | 96.0 | 74.9 | 59.8 | 74.5 | 92.6 | 32.7 | 76.5 |
| HiDeCLIP-ViT-B/32 | 30M | 92.7 | 75.6 | 82.9 | 83.3 | 95.7 | 75.6 | 62.8 | 74.5 | 92.0 | 35.8 | **77.1 (↑ +0.6)** |

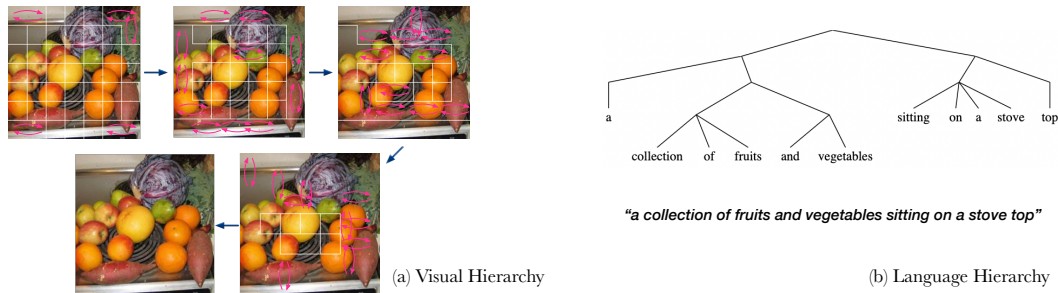

(a) Visual Hierarchy  (b) Language Hierarchy

Figure 4: Visualization results about "a collection of fruits and vegetables sitting on a stove top". Our HiCLIP successfully recognizes the green *vegetable*, *fruits* like the apples as well as the *stove top*. In the mean time, the language hierarchy of the input sentence is also created through analyzing the constituent attention weights.

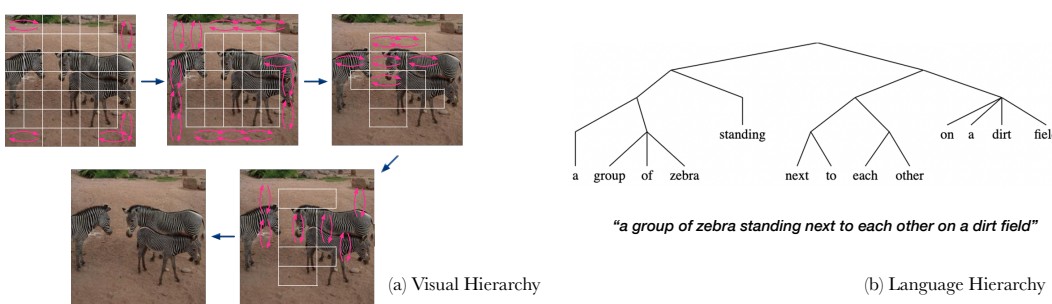

(a) Visual Hierarchy  (b) Language Hierarchy

Figure 5: Visualization results about "a group of zebra standing next to each other on a dirt field". Our HiCLIP approach can generate correct parsing tree while aggregating image patches that correspond to the concepts *zebra* and *dirt field* into common groups in an unsupervised manner.

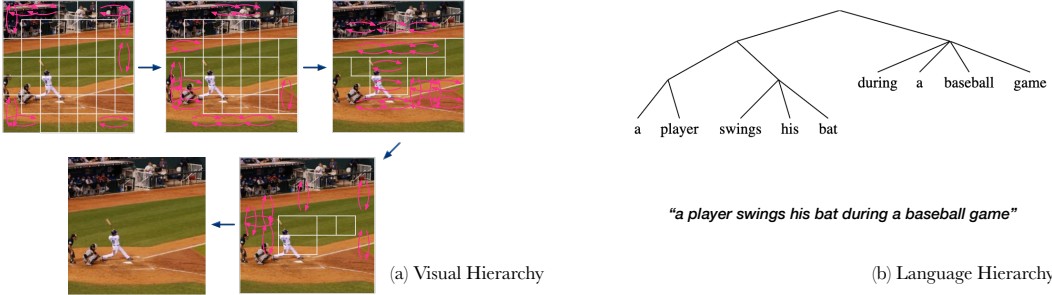

(a) Visual Hierarchy        (b) Language Hierarchy

Figure 6: Visualization results about "a player swings his bat during a baseball game". Our HiCLIP approach can successfully aggregate the regions of dugout and *baseball* field, while the batter is not well recognized in the visual hierarchy. Meanwhile, the language parsing tree is generally correct by analyzing the constituent attention weights.

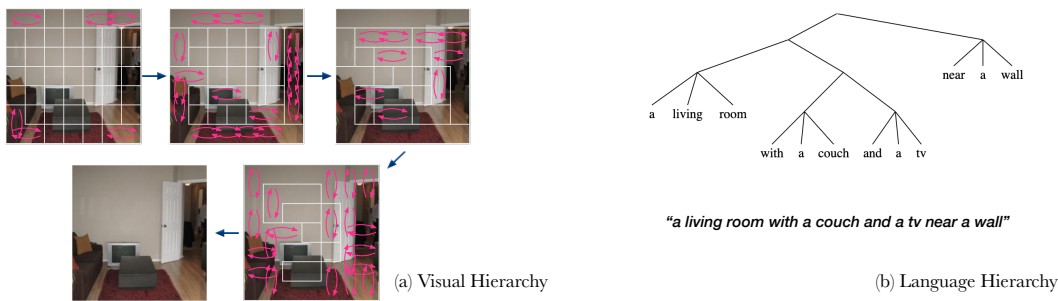

(a) Visual Hierarchy        (b) Language Hierarchy

Figure 7: Visualization results about "a living room with a couch and a tv near a wall". Our HiCLIP successfully generates a correct language hierarchy of the inputs sentence. Moreover, it merges the patches that correspond to the *couch* and *tv*, as well as the carpet and door regions of the *living room*.

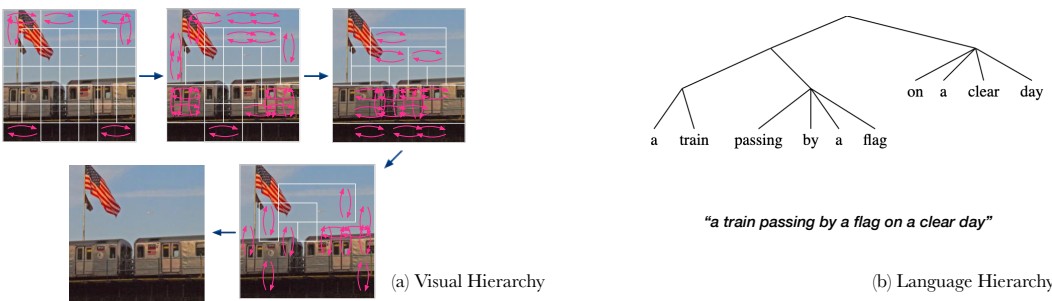

(a) Visual Hierarchy        (b) Language Hierarchy

Figure 8: Visualization results about "a train passing by a flag on a clear day". Our HiCLIP can successfully recognize the regions of *train*, *flag*, and elevated track. For the language hierarchy, HiCLIP can also aggregate these concept words together correctly.

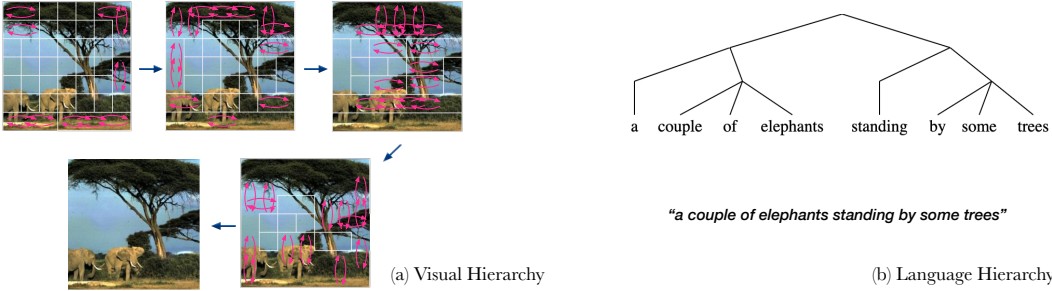

(a) Visual Hierarchy          (b) Language Hierarchy

Figure 9: Visualization results about "a couple of elephants standing by some trees". Our HiCLIP captures a correct language hierarchy of the verb and concept words in the input sentences. Meanwhile, HiCLIP can also aggregate image patches that correspond to the concepts *elephant* and *tree*.

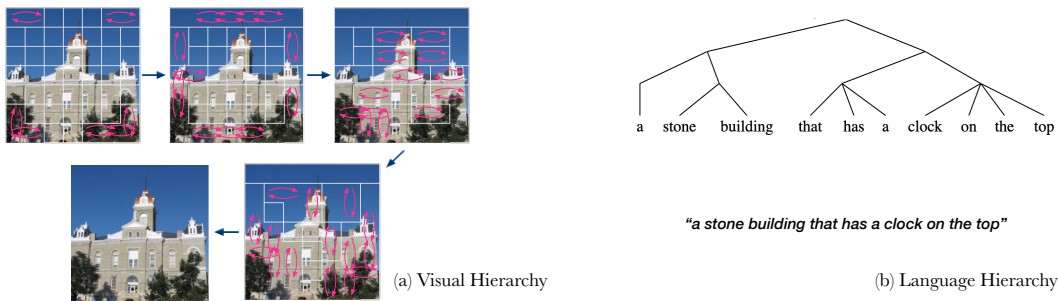

(a) Visual Hierarchy          (b) Language Hierarchy

Figure 10: Visualization results about "a stone building that has a clock on the top". In this case, our HiCLIP can roughly merge the regions of *stone building* and *clock* tower. For the language hierarchy, it seems that "a" and "clock" should be aggregated together first and before "has".

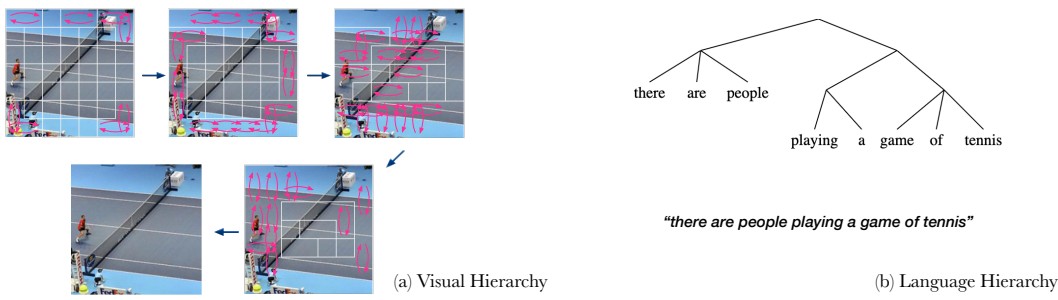

(a) Visual Hierarchy          (b) Language Hierarchy

Figure 11: Visualization results about "there are people playing a game of tennis". In this case, our HiCLIP didn't achieve meaningful visual hierarchy result for the *tennis* court, even though the patches correspond to the player has been merged during the induction process. For the language hierarchy, it seems that "a" and "game" should be aggregated together first and before "playing".

---

**Algorithm 1** Unsupervised hierarchy induction for input images

---

**Require:** All neighboring affinity scores $a^l_{(i,j),(i',j')}$ for $l = \{1, \ldots, N\}$ layers, A list of "break" threshold values $\{\theta_1, \ldots, \theta_N\}$ for every layer.

1: $l \leftarrow N$                                                                    ▷ Start from the highest layer
2: Initialize a nested list $\mathcal{B} = \{B_1, \ldots, B_N\}$                        ▷ Store break edges of each layer
3: **while** $l > 0$ **do**
4:    **for each** edge $(i,j), (i',j')$ in the patch graph **do**
5:        **if** $a^l_{(i,j),(i',j')} < \theta_l$ **then**
6:            **if** $l = N$ **then**
7:                Append the edge $(i,j), (i',j')$ to $B_l$        ▷ Break the edge $(i,j), (i',j')$ in the top layer $N$
8:            **else**
9:                **if** edge $(i,j), (i',j')$ not in $B_{l+1}$ **then**
10:                   Append the edge $(i,j), (i',j')$ to $B_l$         ▷ Break the edge $(i,j), (i',j')$ in layer $l$
11:               **end if**
12:           **end if**
13:       **end if**
14:   **end for**
15:   $l \leftarrow l - 1$                                                               ▷ Move to the next lower layer
16: **end while**
17: Draw visual hierarchy based on $\mathcal{B}$, then remove redundant edges by finding connected components.

---

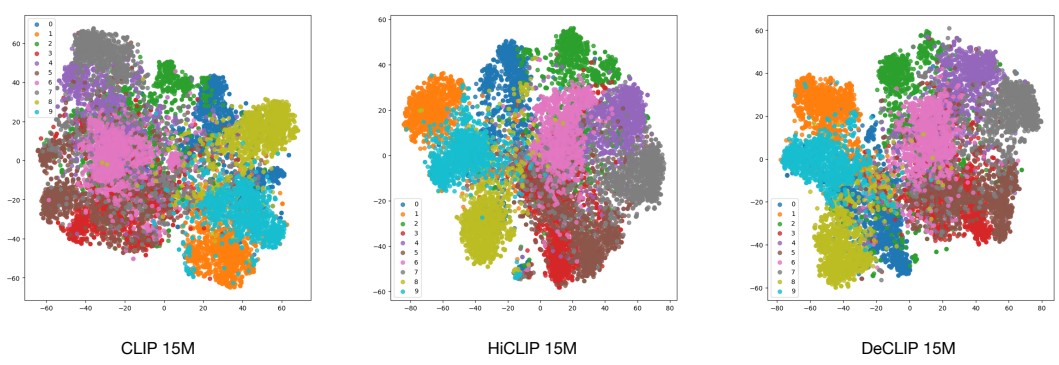

Figure 12: Visualization of the learned feature space via t-SNE on CIFAR-10 dataset. We use CLIP, HiCLIP, DeCLIP checkpoints that pretrained on *YFCC-15M* data.

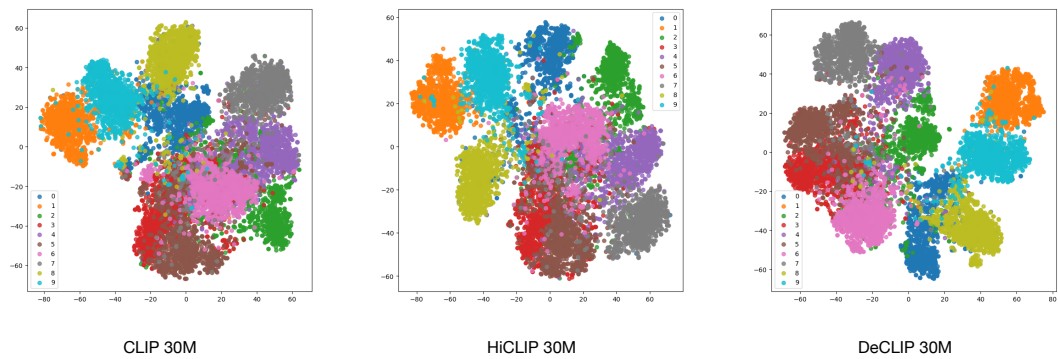

Figure 13: Visualization of learned feature space via t-SNE on CIFAR-10 dataset. We use CLIP, HiCLIP, DeCLIP checkpoints that pretrained on *30M* data.

Layer 0:

$$a^0_{0,1} = 0.40 \quad a^0_{1,2} = 0.66 \quad a^0_{2,3} = 0.24 \quad a^0_{3,4} = 0.71 \quad a^0_{4,5} = 0.34$$

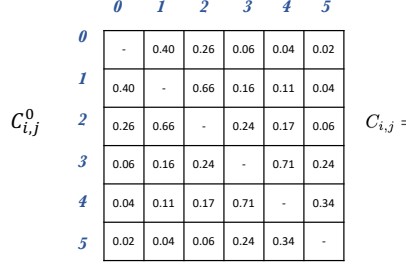

$C^0_{i,j}$

|   | 0 | 1 | 2 | 3 | 4 | 5 |
|---|---|---|---|---|---|---|
| 0 | - | 0.40 | 0.26 | 0.06 | 0.04 | 0.02 |
| 1 | 0.40 | - | 0.66 | 0.16 | 0.11 | 0.04 |
| 2 | 0.26 | 0.66 | - | 0.24 | 0.17 | 0.06 |
| 3 | 0.06 | 0.16 | 0.24 | - | 0.71 | 0.24 |
| 4 | 0.04 | 0.11 | 0.17 | 0.71 | - | 0.34 |
| 5 | 0.02 | 0.04 | 0.06 | 0.24 | 0.34 | - |

$$C_{i,j} = \prod_{k=i}^{j-1} a_{k,k+1}$$

Layer 1:

$$a^1_{0,1} = 0.43 \quad a^1_{1,2} = \mathbf{0.99} \quad a^1_{2,3} = 0.25 \quad a^1_{3,4} = \mathbf{0.99} \quad a^1_{4,5} = 0.54$$

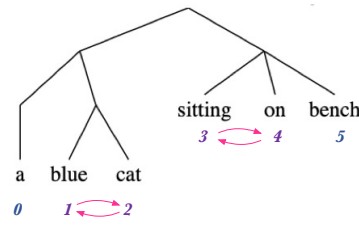

$C^1_{i,j}$

|   | 0 | 1 | 2 | 3 | 4 | 5 |
|---|---|---|---|---|---|---|
| 0 | - | 0.43 | 0.43 | 0.11 | 0.11 | 0.06 |
| 1 | 0.43 | - | 0.99 | 0.25 | 0.24 | 0.13 |
| 2 | 0.43 | 0.99 | - | 0.25 | 0.25 | 0.13 |
| 3 | 0.11 | 0.25 | 0.25 | - | 0.99 | 0.54 |
| 4 | 0.11 | 0.24 | 0.25 | 0.99 | - | 0.54 |
| 5 | 0.06 | 0.13 | 0.13 | 0.54 | 0.54 | - |

$$C_{i,j} = \prod_{k=i}^{j-1} a_{k,k+1}$$

Layer 2:

$$a^2_{0,1} = 0.79 \quad a^2_{1,2} = \mathbf{1.00} \quad a^2_{2,3} = 0.32 \quad a^2_{3,4} = \mathbf{0.99} \quad a^2_{4,5} = \mathbf{0.90}$$

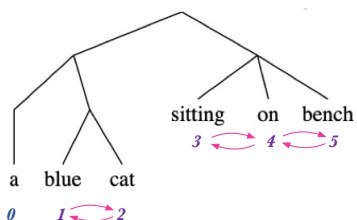

$C^2_{i,j}$

|   | 0 | 1 | 2 | 3 | 4 | 5 |
|---|---|---|---|---|---|---|
| 0 | - | 0.79 | 0.79 | 0.25 | 0.25 | 0.23 |
| 1 | 0.79 | - | 1.00 | 0.32 | 0.32 | 0.29 |
| 2 | 0.79 | 1.00 | - | 0.32 | 0.32 | 0.29 |
| 3 | 0.25 | 0.32 | 0.32 | - | 0.99 | 0.89 |
| 4 | 0.25 | 0.32 | 0.32 | 0.99 | - | 0.90 |
| 5 | 0.23 | 0.29 | 0.29 | 0.89 | 0.90 | - |

$$C_{i,j} = \prod_{k=i}^{j-1} a_{k,k+1}$$

Figure 14: Detailed illustration of the computation of $C$. We take a short sentence "a blue cat sitting on bench" as a toy example. We show real values of $a^l_{i,i+1}$ and calculated $C^l_{i,j}$ matrices in the first three layers of Tree-Transformer. We can clearly see that several words are grouped together when the affinity score between them is high enough and greater than a threshold (e.g., 0.8): *blue* and *cat* in Layer 1; *sitting* and *on* in Layer 1; *sitting*, *on*, and *bench* in Layer 2.

