# OpenReview forum: "HiCLIP: Contrastive Language-Image Pretraining with Hierarchy-aware Attention"
_ICLR.cc/2023/Conference — ICLR 2023 poster_

### Official Review · Reviewer_79pR · 2022-10-23

**Confidence:** 4
**Correctness:** 4
**Technical Novelty And Significance:** 3
**Empirical Novelty And Significance:** 3
**Recommendation:** 6

**Clarity, Quality, Novelty And Reproducibility:**

Clarity: Overall the paper writing is easy to follow. But I still have some questions regarding some details:
- I cannot find the definition of $a_{i,i+1}$ but only $\hat{a}_{i,i+1}$.
- Not clear about the visualization process, I cannot find it on the main paper or supplemental materials.

Quality, Novelty: c.f. Pros & Cons

Reproducibility:
Will the authors release the codes? I cannot find any code implementations in the supplemental materials.

**Details Of Ethics Concerns:**

No ethics concerns for me.

**Strength And Weaknesses:**

Pros:
- The idea of exploring vision and language hierarchical representations for better VL pretraining is soundness and reasonable.
- The method of the group transformer on images is a novel extension of the tree transformer.
-  The experiments results show a large improvement compared with CLIP baselines.

Cons:
- The paper claims to build language structures like unsupervised grammar induction (also in visulization). I'm wondering about the qualitative results on grammar induction (e.g. on COCO dataset like V-PCFG and CLIORA).
- Computing the hierarchy-aware mask seems complicated and time-consuming. What is the inference time comparing CLIP-based methods?
- Eq. 9 computes the merging score for two patches by connecting the points along the grid with only one turn. Are there any arguments that use these two paths rather than some other paths like zig-zag?
- Lack of in-depth analysis on the method though the effectiveness of G-trans and T-trans are well verified in the ablation study. In other words, I would like the authors to discuss the motivation behind the model design and how it works.

Suggestion:
- The paper explores VL structure for better image-text alignment, but ignores region-token alignment (in FILIP) and hierarchical alignment (in CLIORA). I hope the authors can discuss the possibility of extending the current work to the fine-grained cross-modal alignment as mentioned in abstract. This is not a weakness so it won't affect the final score.

**Summary Of The Paper:**

This paper develops a hierarchy-aware attention mechanism for vision-language pretraining (CLIP-based) models. Motivated by the observation that both vision and language have structural representation, this paper aims to group the similar concepts in a hierarchical manner. It adopts a tree transformer to encode the language structure and designs a novel group transformer to encode image structure. Experimental results show a significant improvement compared to CLIP-based baselines.

**Summary Of The Review:**

I like the idea of exploring the hierarchical representation (in a weakly or unsupervised manner) for VL understanding. This paper presents an interesting design to incorporate VL structural information to representation learning. My overall reviews are positive, and I'm looking forward to more discussions with the authors about my concerns.

---

> ### Author Response · Authors · 2022-11-17
> **For Reviewer 79pR**
>
> Dear reviewer:
>
> Thank you for your helpful comments and suggestions. Here are our responses to your questions:
>
> > Q1:  I'm wondering about the qualitative results on grammar induction.
>
> A1: We conducted grammar induction on the test split of MSCOCO caption and used the ground-truth provided by the VG-NSL [1] paper. The F1 scores of different approaches over all 5,000 captions are:
>
> Baselines:
> Random 27.1, Left 23.3, Right 22.9, PMI 30.5, Gumbel 27.9, ON-LSTM 45.5, VG-NSL 50.4, PRPN 52.5, C-PCFG 53.6, CLIORA 56.2
>
> HiCLIP approaches:
> HiCLIP (ViT-B/32 + 15M) 34.9, HiCLIP (ViT-B/16 + 15M) 37.3, HiCLIP (ViT-B/32 + 30M) 39.2, HiDeCLIP (ViT-B/32 + 15M) 40.9
>
> According to the result, HiCLIP performs noticeably better over the random baseline in terms of the F1 scores indicating that HiCLIP still successfully learns linguistically meaningful hierarchies. However, since HiCLIP only uses image-text global alignment information, our HiCLIP achieved a lower F1 score than the aforementioned research works that are designed for the grammar induction task, while the above visually-guided grammar induction approaches [1] [2] typically take extracted object features as inputs.
>
> From the F1 scores of HiCLIP approaches thanks to your advice, we find that there are potentially a few ways to improve HiCLIP’s performance, namely (1) using finer-grained visual patches or object-level information, and (2) applying additional self-supervised finer-grained training objectives than a global contrastive loss may improve the performance of grammar induction.
>
> [1] Shi, Haoyue, Jiayuan Mao, Kevin Gimpel, and Karen Livescu. "Visually Grounded Neural Syntax Acquisition." In Proceedings of the 57th Annual Meeting of the Association for Computational Linguistics, pp. 1842-1861. 2019.
>
> [2] Wan, Bo, Wenjuan Han, Zilong Zheng, and Tinne Tuytelaars. "Unsupervised Vision-Language Grammar Induction with Shared Structure Modeling." In International Conference on Learning Representations. 2022.
>
>
> > Q2: What is the inference time comparing CLIP-based methods?
>
> A2: For inference time, we evaluate the FLOPs of CLIP, HiCLIP (T-Trans only), HiCLIP (G-Trans only), HiCLIP (using both T-Trans and G-Trans):
>
> CLIP: 7.418G, HiCLIP (T-Trans only): 7.945G, HiCLIP (G-Trans only): 8.15G, HiCLIP (using both T-Trans and G-Trans): 8.677G
> The complete HiCLIP has 15% more time complexity than the original CLIP.
>
> In this work, we keep the same number of tokens across Transformer layers and only compute the soft affinity scores to indicate their grouping together. In the future, our method can be improved by hard merging several old tokens into a new token, thus reducing the computational complexity of HiCLIP ( discussions on this can also be found in our response to Q2 of Reviewer WpY2).
>
>
> > Q3: Eq. 9 computes the merging score for two patches by connecting the points along the grid with only one turn. Are there any arguments that use these two paths rather than some other paths like zig-zag?
>
> A3: In HiCLIP, we chose the one-turn strategy because of two reasons: 1) it is a straightforward approximation of finding the shortest path; 2) we are able to transfer it to tensor operations so as to ensure the hierarchy-aware attention module to be differentiable. Therefore, HiCLIP can be trained in the end-to-end manner, and there is less time complexity bottleneck in computing the attention mask C.
>
> For the zig-zag style path strategy, it is another promising approximation option for the shortest path and thank you for bringing it up. It could be possible to implement it using purely tensor operations the same way as we implemented our one-turn strategy, but we didn’t find such a good solution as for now. If we cannot transfer it into tensor operations, there will be higher overhead in the computation of the attention mask C. We will leave the exploration of other path strategies as our future work and update the manuscript accordingly.

---

> > ### Author Response · Authors · 2022-11-17
> > **For Reviewer 79pR (Cont'd)**
> >
> > > Q4: Lack of in-depth analysis on the method though the effectiveness of G-trans and T-trans are well verified in the ablation study. In other words, I would like the authors to discuss the motivation behind the model design and how it works.
> >
> > A4: We summarize our motivation behind our model design as follows: Like what Reviewer WpY2 points out, there have been substantial research works exploring or discovering visual or linguistic hierarchies to assist model design and improve the model’s capabilities on various vision and language tasks. Since both images and text contain rich semantic hierarchies, we believe that explicitly modeling these hierarchies as hierarchy-aware attention is beneficial for multimodal semantic alignment as well as high-level understanding and reasoning. For understanding how $C$ works, we provide a toy sentence example in Figure 8 of Section E in the Appendix. The figure offers an intuitive example to show how $C$ is computed and how the values in it can be used to aggregate semantically similar tokens, and thus generates a hierarchy. We are working to generate more visualization examples and will update soon.
> >
> >
> > > Q5: The paper explores VL structure for better image-text alignment, but ignores region-token alignment (in FILIP) and hierarchical alignment (in CLIORA). I hope the authors can discuss the possibility of extending the current work to the fine-grained cross-modal alignment as mentioned in the abstract.
> >
> > A5: Thanks for the suggestion. In this work, we focus on improving CLIP with hierarchy-aware attention but keeping the original global-level image-text contrastive learning loss. And we agree that it is feasible and very promising to extend our work for fine-grained cross-modal alignment either: 1) we can take fine-grained alignment loss similar to FILIP’s during training so that our model can perform region-token alignment and potentially achieve better performance; or 2) similar to CLIORA uses pretrained object detector to get bounding box regions with explicit semantic meanings. And we suspect that by combining the fine-grained training objective the performance gain of leveraging hierarchical modeling could be more noticeable. We will add this discussion into our future work and explore this promising direction.
> >
> > > Q6: I cannot find the definition of $a_{i, i+1}$ but only $\hat{a}_{i, i+1}$.
> >
> > A6: According to the non-splittable property, we ensure all layer-wise affinity scores $a^l_{i, i+1}$ increase monotonously as the network goes deeper. So we utilize both the original neighboring affinity score $\hat{a}^l_{i, i+1}$ and the previous layer’s affinity scores $a^{l-1}_{i, i+1}$ to calculate the current layer's affinity scores --
> >
> > $a^l_{i, i+1}$, which is in line with Eq. (5). We update a definition of $a_{i, i+1}$ on the fourth page of the paper PDF.
> >
> >
> > > Q7: Will the authors release the codes?
> >
> > A7: Yes. We will release our code upon acceptance.
> >
> >
> > As we spent much time conducting additional experiments suggested by all reviewers, we are still working on generating more visualization cases. We will update them as well as the algorithm for the visualization process of HiCLIP soon.

---

> > > ### Author Response · Authors · 2022-11-18
> > > **For Reviewer 79pR (Cont'd)**
> > >
> > > Dear reviewer:
> > >
> > > In our updated version PDF, we have provided six more unsupervised induction cases of HiCLIP. Furthermore, according to your advice, in **Section C of the Appendix of our updated PDF**, we elaborate on how we conduct unsupervised hierarchy induction with HiCLIP (i.e., detailed descriptions of the visualization process).

---

### Official Review · Reviewer_jjXQ · 2022-10-24

**Confidence:** 3
**Correctness:** 3
**Technical Novelty And Significance:** 3
**Empirical Novelty And Significance:** 4
**Recommendation:** 6

**Clarity, Quality, Novelty And Reproducibility:**

The experiments are clear and easy to follow, but I thought the approach section (Section 3) could have been restructured a little bit. I think preliminaries (like the tree transformer part) could have been discussed first very explicitly so that it's easy to follow what are the preliminaries and how its extended. I think the figures can also use some more explanation. For instance, in figure 2, what does x mean? What do cyclic arrows represent?

**Strength And Weaknesses:**

Strengths
The paper proposes a way to create compact image and text embeddings that consistently outperforms CLIP style approaches for many tasks like classification, image-text retrieval etc.

Weaknesses
1. The one-turn path taken to compute affinity score between two patches is a pretty rough approximation of shortest path. Is the shortest path really that computationally expensive? I am not sure if shortest path is the right thing to do. Something like connected components (measured by some sort of a threshold) seems like the right thing to do. But concretely, can the author compare these two approaches (the shortest path vs the path with only one turn) and show some empirical result which one does better?

2. Tree transformer in language seems to be more important than Group Transformer on the vision side, even for visual tasks like image classification (Table 4, Row 2 vs Row 3). For smaller models, the trends seems to be reserved (Row 6 vs Row 8). Can the authors explain this trend a little bit more?

3. Why not show 11 datasets average accuracy as well in ablations of Table 4.

4. From the visualizations in Figure 3, 4 and 5, frankly, it's very unclear what the visual heirarchies are capturing. For instance in figure 5, I don't see how patches belonging to zebra and dirt fields are grouped together.

**Summary Of The Paper:**

The paper introduces hierarchy-aware attention in both the visual and textual streams of CLIP. The paper extends the work of 1D Tree transformers and apply it to image patches in 2D.  They show better performance on vision-only and vision-and-language downstream tasks than ClIP style approaches.


**Summary Of The Review:**

Overall, the proposed approach shows empirical improvements over CLIP but apart from that, I did not gain any other significant insights for why / how it works. Some of these gains are coming from the tree-transformer-like hierarchy attention in the text encoder, which dilutes the significance of the more interesting contribution in the paper (the hierarchical attention for 2D visual patches). I am on the fence about this paper, and I will update my thoughts based on the author's feedback and other reviews.

Update post-rebuttal: The rebuttal addresses all my concerns - specifically, the connection to connected components, why not to use shortest-path, and explanation of qualitative figures. After reading all the responses to other authors, I am improving the score to 6 (weak accept).

---

> ### Author Response · Authors · 2022-11-17
> **For Reviewer jjXQ**
>
> Dear reviewer:
>
> Thanks for your comments and advice. We address your questions as below:
>
> > Q1: Is the shortest path really that computationally expensive?
>
> A1: Yes. The current solution is much faster than the shortest path approach.
>
> In fact, we have implemented the shortest path strategy using the Floyd-Warshall algorithm [1]. Since there are so many for loops in the Floyd-Warshall algorithm, we used the Numba library (http://numba.pydata.org/) to accelerate the computation of the shortest path strategy. However, if we consider the batch dimension, we need an O(N^4) time complexity to get the shortest path connecting all patches across a batch of images. As a result, when we pretrain Floyd-version HiCLIP on a 5M data subset over 8 A100 GPUs, it will take 158 hours (6.6 days) to finish the pretraining. In contrast, our current HiCLIP solution only takes 21 hours (~7 times faster). Another reason we don’t choose the shortest path strategy is that it is difficult to be represented as basic tensor operations so that in the current format it is not differentiable, which makes it hard to conduct end-to-end learning. We have converted the current one-turn strategy in tensor operations and the details will be included in our released codebase.
>
> [1] Floyd, Robert W. "Algorithm 97: shortest path." Communications of the ACM 5, no. 6 (1962): 345.
>
> We are still working on the analysis of the comparison with connected components and will post this part of response later.
>
>
> > Q2: Tree transformer in language seems to be more important than Group Transformer on the vision side, even for visual tasks like image classification (Table 4, Row 2 vs Row 3). For smaller models (vitb/16, more tokens), the trends seem to be reserved (Row 6 vs Row 8). Can the authors explain this trend a little bit more?
>
> A2: From Table 4, we can see that when the visual encoder is of coarse granularity (ViT-B/32), the Tree transformer on the language side plays a more important role than the Group Transformer on the vision side. But when we change to a finer-grained visual encoder (ViT-B/16), the Group Transformer on the vision side shows increasing importance. This can also be observed in the 11 dataset average accuracy. Comparing Row2 and Row 4 (both ViT-B/32), adding G-Trans achieves 3.4% improvement, while when we compare Row 6 and Row 8 (both ViT-B/16), adding G-Trans can get a higher 5.3% improvement.
>
>
> > Q3: Show 11 datasets average accuracy as well in ablations of Table 4
>
> A3: Thanks for your advice. We have **added the average accuracy of 11 datasets to Table 4 of the updated paper PDF**.
>
>
> > Q4: Unclear what the visual hierarchies are capturing in Figure 3, 4 and 5
>
> A4: To better illustrate how visual hierarchies and neighborhood aggregation are conducted, in these figures, we add double arrows to indicate which two components are merged/grouped together. With the help of double arrows, it will be easier to understand the merging trends from layer to layer.
>
> For more visualization cases, we will update them to the paper soon.

---

> > ### Author Response · Authors · 2022-11-18
> > **For Reviewer jjXQ (Cont'd)**
> >
> > Dear Reviewer:
> >
> > In our updated version PDF, we have provided six more unsupervised induction cases of HiCLIP. You can also find detailed descriptions of the visualization process in **Section C of the Appendix of our updated PDF**.
> >
> > In addition, we also provide **detailed discussions about connected components (measured by some sort of threshold) and shortest path design (e.g., HiCLIP)**. We think connected components are an interesting idea and we found our shortest path design (or any path) shares many similarities with connected components. In the following, we will:
> >
> > 1. Formulate the definition of connected components on a patch-graph
> > 2. Showing under this definition, our merged image patches are connected components (or equivalently, we will prove: the connected components in our patch graph have high intra-similarity)
> >
> > * Patch graph
> >   * Image patches naturally form a grid-like graph where the nodes are patches, and we add edges to connect each patch with its 4-neighborhood patches.
> >
> > * Definition of connected components on a patch-graph
> >   * As a grid-like patch graph is naturally a connected graph, we define the connected component $X$ on the patch graph as:
> >     * $X$ is a connected subgraph of the patch-graph
> >     * $X$ has high intra-similarity, means if nodes (patches) $i, j \in X$, then sim$(i, j)$ > thre, here thre < 1
> >
> > * CLAIM: HiCLIP groups similar patches as connected components with high intra-similarity
> >   * In the "unsupervised hierarchy induction part" of our HiCLIP paper, We first get the neighborhood similarity, then extend it to all pairs' similarity as $C$. Then we split neighboring patches with low similarity. Such a "split" operation can be seen as we remove edges from the patch graph with neighborhood similarity below a threshold $z$). If we remove more edges, we will divide the patch graph into separately connected subgraphs (or connected components). In the next, we will prove --> these connected components have high intra-similarity
> >   * Given a connected component $X$, for any two patches $i, j$ in $X$:
> >     * $i,j$ are connected --> there is a path $P$ connect $i,j$
> >     * Suppose the length of $P$ (here length means how many patches on $P$) is $N$, then for all $N-1$ neighborhood pairs on $P$, as they are not removed by "split" operation, so the similarity between them are all greater than $z$. Then following this path, $C_{i,j}$, the similarity between $i,j$ is at least $z^{N-1}$ (here at least because there could be more than one path connecting $i,j$), therefore $\lim_{z \to 1} C_{i,j} = 1$. So with a large enough $z$, we know $X$, the connected component has high intra-similarity
> >
> > * Above is an informal sketch proof, however, we believe it is enough to show our connections with connected components.

---

### Official Review · Reviewer_WpY2 · 2022-10-27

**Confidence:** 3
**Correctness:** 3
**Technical Novelty And Significance:** 3
**Empirical Novelty And Significance:** 4
**Recommendation:** 8

**Clarity, Quality, Novelty And Reproducibility:**

**Clarity**
Overall, the paper was easy to read. One very important exception was the second half of Sec 3.1.2: I found it difficult to understand the second portion of HA as noted in the weaknesses. There are also many minor typos and grammatical mistakes which I point out some of them below.

**Reproducibility:**
- What's the batch size?
- Do you plan on releasing code? While the method is explained well, I don't think it would be easy to implement this or reproducing it without access to code or pseudocode of the method.

**Minor suggestions/typos:** I noticed several typos while reading the paper. I list some below, but this list is not comprehensive.
- Abstract: "multimodality content understanding" -> "multimodal content understanding"
- The second sentence in the abstract was very confusing. I encourage the authors to rephrase it.
- Abstract: "CLIP features can hardly reflect the hierarchy nature" -> "CLIP features hardly reflect the hierarchical nature". Furthermore, I think this claim is unsupported. There are several hierarchies that exist in the word. This paper appears to capture the syntactic trees within language and objects masks within images. However, one could imagine a hierarchy over types similar to WordNet (dogs are a superclass of poodle and bulldog) or graph structure depicting scenes (room has chairs and tables) or part-structure (chair has legs). While the paper shows that adding an explicit attentional mechanism to capture one form of structure improves performance, it does not show that CLIP's feature do not capture any hierarchy.
- Sec 3, first paragraph, first line: "share a hierarchy nature" -> "share a hierarchical nature" second line: "The lower level hierarchy" -> "the lower level of the hierarchy"

**Strength And Weaknesses:**

**Strengths**:
- The idea of taking advantage of the hierarchy implicit within visual and language data is very interesting, and it's nice to see such an observation translated to performance improvements.
- the details of implementing attention seem to be well-though of. I specifically liked how the authors explicitly listed the hierarchy aggregation priors.
- I appreciated how the authors went beyond the typical evaluation sets towards some reasoning tasks and analyzed their importance with respect to their modeling contributions.
- The improvements are impressive, especially in cases where HA outperforms the use of SSL tasks in DeCLIP.

**Weaknesses:**
- The missing numbers for SLIP and FILIP in Table 1 are problematic. It seems that the numbers match the DeCLIP github repo, however, both models are capable of doing the 0-shot tasks. SLIP only reported numbers for VIT-B/16 which is different from the backbone used in table 1, so one cannot compare numbers across tables. I want to note that I do not think that SLIP is a very important comparison since it's implicit within DeCLIP, however, FILIP is an important comparison. FILIP does a comparison between tokens, and hence, allows the model to implicitly learn features that allow a mapping between both modality-specific tokens. This is a different way to dealing with the structure between tokens: CLIP aggregates all tokens with equal weight, HICLIP learns hierarchical aggregation, FILIP aggregates the loss/similarity through basically though a form of "cross-modal attention." Understanding how it performs would be helpful to understand what aggregation matters for training. Finally, since the paper reports numbers on ViT-B/16 in table 3, evaluating the pre-trained FILIP weights from DeCLIP should be easy as they seem to have the same training setup.
- The authors do not mention GroupViT by Xu et al. (CVPR 2022). This paper changes the aggregation function through a learned tokenization. As far as I understand, their aggregation is also non-splittable, however, unlike HA-Attention, their grouping block is global rather than local and their computation of affinities are different. GroupViT also uses the same training task and loss as CLIP, so there's a lot of overlap. I think it would be very important for this paper to discuss GroupViT as well as SlotAttention by Locatello et al (NeurIPS 2020). There has been some additional work in that area, so there might be additional work to compare to that I am not aware of, but I think GroupViT is an important comparison and SlotAttention and GroupViT are both important works to discuss as they propose augmentations to attention or transformers with the goal of understanding the composition of the scene to improve learning.
- The paper makes many claims about how previous approaches in vision overlook hierarchies (first line of second paragraph in intro), however, this is not true. First, convolutional networks were initially proposed to capture hierarchies of features through progressively larger receptive fields. This was shown by many papers and notably discussed by Olah et al (2017). Furthermore, vision datasets are often collected via hierarchies with ImageNet collected using WordNet. Pretrained models also often implicitly learn this hierarchy which can show in confusion patterns as shown by Alsallakh et al (2017). Work on scene graphs and scene compositions defines problems that tries to very explicitly capture different hierarchies. Finally, the paper used to discuss how humans perceive the world hierarchically (Kuzovkin et al) compares brain activations with neural network activations and show they correlate. I do not think any of this requires comparison, but the current tone of the introduction greatly overclaims how hierarchy has been overlooked within vision and multimodal research.
- While I liked the idea of showing examples of hierarchy in Fig 3, 4, 5 ... I think it would be nice to show more. I am also not sure if there's a way to quantitatively evaluate this via segmentation (similar to GroupViT) or comparing to parse trees for language. This is more of a suggestion than a weakness.
- I find it very difficult to understand how C is used or computed. Here's my understanding and points I found confusing. I would greatly appreciate some clarification and I think the paper would greatly benefit from an illustrated example of those values (or at least just C) for a small graph (C for the hierarchy of a 4 or 5 word sentence as a toy example would be sufficient I think)
	- For N tokens, you compute affinity values. I am assuming equation 3 can be used for $s_{i, i-1}$ otherwise, it's unclear how one gets those values for equation 4. Furthermore, it's unclear how to compute the similarity for edge tokens; I can imagine using start and end tokens in language, but how do you do that for image tokens?
	- Equation 5 shows how affinities are updated between layers to ensure that affinities are monotonically non-decreasing, however, that doesn't mean that the order of affinities will not change with time (eg, $a_{0,1} = 0.5 \text{ and }a_{1,2} = 0.4$ at $t_0$ but , $a_{0,1} = 0.55 \text{ and }a_{1,2} = 0.6$ at $t_1$). How would that affect the model? is there something that I am missing that prohibits that from happening? If the ordering changes, does that affect the implied hierarchy.
	- Most importantly C appears to only be computed for $j>i$ , is this true? if not, I think you need to update the notation to explain that as well as explain how to set the values $a_{i,i}$. Additionally, C appears to be a set of values between 0 and 1 which is multiplied by the QK component of the attention. It does not appear to create a strict hierarchy with definitive edges that only connect parents and children.


 **References:**
- Xu, J., De Mello, S., Liu, S., Byeon, W., Breuel, T., Kautz, J., & Wang, X. (2022). GroupViT: Semantic Segmentation Emerges from Text Supervision. In _Proceedings of the IEEE/CVF Conference on Computer Vision and Pattern Recognition_ (pp. 18134-18144).
- Locatello, F., Weissenborn, D., Unterthiner, T., Mahendran, A., Heigold, G., Uszkoreit, J., ... & Kipf, T. (2020). Object-centric learning with slot attention. _Advances in Neural Information Processing Systems_, _33_, 11525-11538.
- Olah, et al., "Feature Visualization", Distill, 2017. https://distill.pub/2017/feature-visualization/
- Bilal, Alsallakh, et al. "Do convolutional neural networks learn class hierarchy?." _IEEE transactions on visualization and computer graphics_ 24.1 (2017): 152-162.



**Summary Of The Paper:**

The paper makes the observation that classic attention mechanisms do not take advantage in the hierarchical structure implicit within language and vision.
The propose augmenting the attention operators within transformers with a hierarchy-aware module that progressively discovers "semantic hierarchies" between the tokens.
The new module is applied within the scope of vision and language pre-training (CLIP) and evaluated primarily on CLIP and DeCLIP (a method that combines CLIP with several other self-supervised tasks) with some comparisons to SLIP (CLIP + SimCLR) and FILIP (A variant of CLIP that computes image-caption similarity on the tokens instead of cosine similarity between global vectors).
The proposed attention can improve performance of both CLIP and DeCLIP on downstream visual discrimination and vision and language tasks. Furthermore, augmenting CLIP with hierarchical attention can sometimes improve performance more than adding other SSL tasks; HiCLIP > DeCLIP for some vision and language tasks.
Furthermore, a few qualitative examples are provided that show that the learned hierarchy is meaningful. Several ablations are conducted to evaluate the impact of dataset size (more data helps), backbone parameters (smaller patches resulted in performance improvements), and hierarchy on each domain (hierarchy-attention seems to benefit language encoders more than visual encoders, and using both helps even more).

**Summary Of The Review:**

Overall, I think the paper presents an interesting idea with a good motivation.  The results are compelling and show strong improvements over prior methods. However, I found the explanation of how hierarchical attention operated very confusing, with some unclear notation. Furthermore, there are some missing comparisons. I tentatively set the rating as marginally above acceptance, but I am happy to raise my rating if the points below are adequately addressed.

Below, I summarize the major weaknesses and clarifications in order of importance that I hope the authors will address:
- The hierarchical attention explanation is a bit confusing, especially how C is computed and used. I would appreciate some clarification in the discussion and I believe the paper would greatly benefit from some additional explanation and a toy example clarifying how C is computed and used.
- A discussion of GroupViT and SlotAttention would be important to add, as well as a comparison to GroupViT as an alternative approach for aggregating information and a relevant baseline.
- Table 1 is incomplete for no clear reason. As I noted, there are published numbers that share the setup using a ViT B/16 which was trained by the authors are reported in Table 3. While the comparison against SLIP is not crucial, I think comparing against FILIP and DeFILIP would be very helpful.
- I think the claims about hierarchy being overlooked in computer vision and multimodal representation learning should be adjusted or better contextualized.

----------------------
**Update (Nov 17th):** I updated my recommendation from 6 to 8 as the authors have responded to my major concerns.

---

> ### Author Response · Authors · 2022-11-17
> **For Reviewer WpY2**
>
> Dear reviewer:
>
> Thank you for your constructive feedback and we apologize for the delay in response. We were waiting for the results of the GroupViT as suggested so that we could respond with detailed numbers and reply to the aforementioned questions all at once. It did take us some time to figure out the resources and to make sure GroupViT converged. We truly appreciate your comments to help make our paper better!
>
> We provide the following responses to your questions and concerns. Besides, we have corrected the typos and modified our claims in the abstract & introduction in the main paper and will keep proofreading.
>
> > Q1: Comparison with SLIP, FILIP, and DeFILIP would be very helpful.
>
> A1: We ran zero-shot image classification experiments with the open-source checkpoints of SLIP, FILIP, and DeFILIP listed in https://github.com/Sense-GVT/DeCLIP#clip-benchmark. They are all pretrained on the same 15M version data. We **report the performances of SLIP, FILIP, and DeFILIP in Table 1 of the updated paper PDF**. According to the table, SLIP and FILIP achieve average zero-shot classification accuracies of 30.0% and 37.2% over the 11 datasets, respectively, while our HiCLIP has 41.8%. DeFILIP achieves an average zero-shot classification accuracy of 44.2% over the 11 datasets, while our HiDeCLIP is 44.6%.
>
>
> > Q2: Claims about hierarchy being overlooked.
>
> A2: Thanks for pointing out this issue. We appreciate your advice and revise our claim in the abstract, introduction, and related work section accordingly. Please also refer to the updated version PDF (highlighted in blue color).
>
> For your comment about our claim in the abstract, we agree that given large pretraining data, CLIP can possibly capture hierarchies implicitly in a data-driven manner. So we refine our claim to emphasize that CLIP does not "explicitly" capture the hierarchical nature of high-level and fine-grained semantics.
>
> For your comment about our claim in the introduction section, our original claim about the utilization of hierarchical nature is more about utilizing both semantic and spatial hierarchies. However, we agree with your opinion that our current tone of the introduction does not properly summarize the progress in vision hierarchy-related studies. We revised our claims and focused on how CLIP family approaches may have overlooked explicit model hierarchies through merging from local to global for either patches/tokens (hard style) or affinity scores in the hierarchy attention mask (soft style). In addition, we revised the section "Hierarchical Discovery in Vision" in the related work to include the scene graph methods as well as a few other recent studies on vision hierarchies following your suggestions:
>
> While our work explores the soft hierarchy modeling (with a soft attention mask strategy), we also find some recent works that share similar motivations with our HiCLIP. For example, Token Merging (ToMe) [1] and Token Pooling [2]. These two works are built upon another way of extracting hierarchy in vision by merging tokens gradually (in a hard merging manner). The difference between our HiCLIP and the two approaches can be summarized as follows: ToMe and Token Pooling adopt hard token merging to improve the speed of ViT while only suffering from a slight performance drop caused by information loss during merging. Our HiCLIP merges the soft attention mask of tokens to construct a hierarchy, which keeps all information of tokens focusing on a better performance on downstream tasks. While enjoying a performance boost, HiCLIP has 15% more time complexity (in terms of FLOPs) than the original CLIP. As a comparison, ToMe achieves 2-3x faster inference speed up over ViT. It is a natural and promising idea to combine soft and hard token merging in multimodal learning (possibly in different stages/layers) to achieve a better speed-accuracy trade-off. We will leave this as our future work. In terms of CNN and WordNet, we think CNN adopts more inductive bias about spatial hierarchies, while WordNet is collected more at the label level, therefore uses more semantic hierarchies in the word embedding space. The same discussion has been updated in the manuscript.
>
>
> [1] Bolya, Daniel, Cheng-Yang Fu, Xiaoliang Dai, Peizhao Zhang, Christoph Feichtenhofer, and Judy Hoffman. "Token Merging: Your ViT But Faster." arXiv preprint arXiv:2210.09461 (2022).
>
> [2] Marin, Dmitrii, Jen-Hao Rick Chang, Anurag Ranjan, Anish Prabhu, Mohammad Rastegari, and Oncel Tuzel. "Token pooling in vision transformers." arXiv preprint arXiv:2110.03860 (2021).

---

> > ### Author Response · Authors · 2022-11-17
> > **For Reviewer WpY2 (Cont'd)**
> >
> > > Q3: Provide additional explanation and a toy example clarifying how C is computed and used.
> >
> > A3: In our paper, we get the merging tendency C for any image patch pairs by following two steps (here "any" means we include pairs that are/are not spatial neighbors)
> > * We calculate the merging tendency for neighboring patch pairs first.
> > * For non-neighboring patch pairs, their merging tendencies are defined through a one-turn path. On this path, the consecutive neighboring merging tendencies are multiplied (so the final score will be small if one of them is small) as the final merging tendency of this non-neighboring pair.
> > * This means two non-neighboring patches are similar, if and only if: 1) There is a one-turn path connecting them, and 2) On this path, all neighboring pairs are similar.
> >
> > We also have provided a toy sentence example with 6 words to show how $C$ is computed from lower layers to higher layers. Please **refer to Figure 8 of Section E in the Appendix in the updated version PDF**. In this figure, we offer an intuitive example to show how $C^l_{i,j}$ is computed from layer-wise affinity scores $a^l_{i, i+1}$. In addition, we show how the values in $C$ can be used to merge semantically similar tokens, and thus generate a hierarchy progressively.
> >
> > For your other questions about $C$ in the review comment, we address as below:
> >
> > > 1) For N tokens, you compute affinity values. I am assuming equation 3 can be used for $s_{i, i-1}$?
> >
> > A: Yes. It can be used for $s_{i, i-1}$.
> >
> > > 2) Furthermore, it's unclear how to compute the similarity for edge tokens; I can imagine using start and end tokens in language, but how do you do that for image tokens?
> >
> > A: We will only calculate the affinity scores (similarities) between adjacent tokens. For example, if we have 5 input text tokens, then we will calculate four affinity scores ($a_{0,1}$, $a_{1,2}$, $a_{2,3}$, $a_{3,4}$). For an input $3 \times 3$ visual patches, we will have 12 affinity scores ( $2 \times 3 \times (3-1)$ ) between all patches following the four-adjacent neighborhood.
> >
> > > 3) Equation 5 shows how affinities are updated between layers to ensure that affinities are monotonically non-decreasing, however, that doesn't mean that the order of affinities will not change with time. How would that affect the model? If the ordering changes, does that affect the implied hierarchy?
> >
> > A: During training, the attention mask $C$ is updated in a bottom-up style. When conducting unsupervised hierarchy induction, we adopt a top-down manner to gradually break the connections between different visual patches and text tokens. It is possible that the order of affinities can change with time, but since the unsupervised hierarchy induction is performed from higher layers to lower layers, our hierarchy induction method will decide based on the most recent affinity scores. Thus, the ordering changes will not affect the implied hierarchy.
> >
> > > 4) Most importantly C appears to only be computed for $j>i$, is this true?
> >
> > A: In fact, $C$ is a symmetric matrix, so we have $C_{i,j} = C_{j,i}$. We only need to calculate the case for $j>i$. We have added this note to the fifth page of the paper PDF.
> >
> >
> > > 5) It does not appear to create a strict hierarchy with definitive edges that only connect parents and children.
> >
> > A: Yes. The calculated $C$ is a soft attention mask, so this is not a hard/strict hierarchy constraint. However, it is necessary to keep as much information as possible to facilitate understanding, retrieval tasks, and reasoning tasks. In order to get explicit hierarchy from $C$, we set thresholds to decide whether to break adjacent visual patches or text tokens and conduct unsupervised hierarchy induction in a top-down manner.

---

> > > ### Author Response · Authors · 2022-11-17
> > > **For Reviewer WpY2 (Cont'd)**
> > >
> > > > Q4: A discussion of GroupViT and SlotAttention would be important to add, as well as a comparison to GroupViT as an alternative approach.
> > >
> > > A4: We took some time to adapt GroupViT’s model to our framework. To compare fairly with GroupViT, we use our image classification benchmark setting, under which we pretrained GroupViT on YFCC-15M with ViT-B/32 visual encoder size. Since the original GroupViT adopts multi-label contrastive loss, for a fair comparison, we use the original CLIP loss when training GroupViT. We used the learning rate setting listed in the GroupViT codebase and made sure it can converge well through proper warm-up. As a result, GroupViT achieves 29.3% accuracy on ImageNet, while 27.8% average accuracy over all 11 downstream datasets. Both of the performances are lower than our HiCLIP approach (40.5% on ImageNet and 41.8% on 11 downstream datasets). Given these current results, we guess the same parameter reported in the GroupViT paper may not be optimal for this use case, and we are still trying to search for a better parameter set. In addition, we also provide a discussion with GroupViT. Here is a table to compare the difference between GroupViT and HiCLIP:
> > >
> > > |  | Tendency to merge | Non-splittable | Properties |
> > > | :-----| :---- | :---- |  :---- |
> > > | GroupViT | semantically similar | Yes, through token merging | 1) Merging through group embeddings, which are predefined and have fixed number; 2) Merging by semantic similarity, so two far-away image patches could be merged if they are similar; 3) Image branch only |
> > > | HiCLIP | Both spatially and semantically similar | Yes, through increasing nearby token similarity | 1) No group (or any other name) embeddings serve as anchors, allowing more arbitrary merging; 2) Merging by considering both spatially and semantically similar patches, so one patch will merge to its nearby similar patch first, then gradually extend to far-away patches; 3) Both image branch and text branch |
> > >
> > > GroupViT shares similarities with SlotAttention. SlotAttention also employs K slot embeddings to map input features into a predefined and fixed number of categories. To achieve unsupervised object discovery, SlotAttention uses an autoencoder architecture and a reconstruction loss as the self-supervised learning training objective. A unique feature of SlotAttention is that it uses a recurrent module to iteratively update the slot attention map with predefined T steps. According to the GroupViT paper, GroupViT can be regarded as an improved version of the single iteration SlotAttention mechanism. Different from GroupViT and SlotAttention, our HiCLIP doesn't use group embeddings as anchors so it allows more arbitrary merging through soft attention mask $C$.

---

> > > > ### Author Response · Authors · 2022-11-17
> > > > **For Reviewer WpY2 (Cont'd)**
> > > >
> > > > > Q5: I am also not sure if there's a way to quantitatively evaluate this via segmentation (similar to GroupViT) or comparing to parse trees for language. This is more of a suggestion than a weakness.
> > > >
> > > > A5:  For grammar induction (parsing tree generation) in the language part, we conducted grammar induction on the test split of MSCOCO caption and used the ground-truth provided by the VG-NSL [1] paper. The F1 scores of different approaches over all 5,000 captions are:
> > > >
> > > > Baselines:
> > > > Random 27.1, Left 23.3, Right 22.9, PMI 30.5, Gumbel 27.9, ON-LSTM 45.5, VG-NSL 50.4, PRPN 52.5, C-PCFG 53.6, CLIORA 56.2
> > > >
> > > > HiCLIP approaches:
> > > > HiCLIP (ViT-B/32 + 15M) 34.9, HiCLIP (ViT-B/16 + 15M) 37.3, HiCLIP (ViT-B/32 + 30M) 39.2, HiDeCLIP (ViT-B/32 + 15M) 40.9
> > > >
> > > > According to the result, our HiCLIP achieved a lower F1 score than baseline research works that are designed for the grammar induction task. However, HiCLIP only uses image-text global alignment information, while above visually-guided grammar induction approaches typically take extracted object features as inputs. Since the F1 scores of HiCLIP approaches have a large gap over the random baseline, demonstrating that HiCLIP really learns linguistically meaningful hierarchies.
> > > >
> > > > [1] Shi, Haoyue, Jiayuan Mao, Kevin Gimpel, and Karen Livescu. "Visually Grounded Neural Syntax Acquisition." In Proceedings of the 57th Annual Meeting of the Association for Computational Linguistics, pp. 1842-1861. 2019.
> > > >
> > > > As for segmentation, there are two reasons why our current HiCLIP approach cannot directly perform the task: 1) As we discussed in Q4, GroupViT adopts additional grouping embeddings and multi-label contrastive loss in order to get a fixed number of groups/segmentation regions as outputs. In our work, we only utilize the original CLIP loss and apply a soft attention mask $C$ to increase the affinity score of nearby semantically similar tokens. It is non-trivial to directly achieve hard grouping results from soft attention masks. However, if we extend HiCLIP with some finer-grained contrastive training loss or conduct hard token merge, HiCLIP will have the ability to conduct segmentation; 2) The goal of grammar induction is to generate a parsing tree consisting of multiple levels. From the parsing tree alone, we can recover how different tokens were aggregated into a complete sentence. The visual hierarchy generated by HiCLIP is also a multiple-level tree structure. However, segmentation is just one level of the dynamic merge process. It is hard to elicit a level directly as a segmentation result. We think token merge is a promising direction to extend HiCLIP for semantic segmentation and we will leave this as future work.
> > > >
> > > >
> > > > > Q6: What's the batch size?
> > > >
> > > > A6: For the 15m version pretraining data, we set the batch size to 4096 and run all experiments on 32 A100 GPUs. For 30m version pretraining data, we set the batch size to 8192 and run all experiments on 64 A100 GPUs.
> > > >
> > > >
> > > > > Q7: Do you plan on releasing code?
> > > >
> > > > A7: Yes. We will release our code upon acceptance.
> > > >
> > > >
> > > > We are still working on generating more visualization cases. We will update them to the paper PDF soon.

---

> > ### Comment · Reviewer_WpY2 · 2022-11-17
> > **Reviewer Response**
> >
> > Thank you for engaging with the review and for your very thorough response. I will respond here to all your rebuttals and will update my initial review accordingly.
> >
> >
> > Q1. Thanks for including the updated comparison. It is interesting to see the consistent improvement of doing better aggregation where FILIP results in some improvement due to its "cross-modal" attention loss, while HiCLIP results in even better improvements through using the hierarchical structure to bias the feature learning within each encoder.
> >
> > Q2. It was great to see you revise the tone of the introduction. I really liked your proposed idea and discussion, and I think the revised introduction contextualizes it better with respect to prior work.
> >
> > Q3: Thanks for the clarification. I still have one question that I include at the end.
> >
> > Q4: Thanks for including the additional experiments. It is expected that GroupViT will achieve a comparable performance to CLIP on classification as reported in their supplementary material. One difference between your reimplementation (from your discussion) and their method is the patch size, which might be very impactful to the learning process. The more interesting comparison to me would be comparing the groupings/segmentations learned by GroupViT vs. HiCLIP. However, I understand that GroupViT has multiple components to encourage better segmentations, so it's difficult to have a fair comparison. It would be a cool direction for future work.
> >
> > Q5/6/7: thanks for the clarification.

---

> > > ### Author Response · Authors · 2022-11-18
> > > **Reply to Reviewer WpY2**
> > >
> > > Dear Reviewer:
> > >
> > > Thank you for your appreciation of our work as well as providing suggestions that make the paper better.
> > > We will keep revising Section 3.1 to make it easier to follow. Meanwhile, we will provide more implementation details in the Appendix for readers who want a better understanding. In addition, we agree with you that different patch sizes may have a great impact on GroupViT, and we will add this part to the future work discussion as you suggested.
> > >
> > > > Q3: Thanks for the clarification. *I still have one question that I include at the end.*
> > >
> > > We are not very sure if the question is about the clarity of hierarchy attention mask $C$'s computation in Section 3.1.2. Could you tell us which specific question you are referring to?

---

> > > > ### Comment · Reviewer_WpY2 · 2022-11-18
> > > > **Response**
> > > >
> > > > Thank you for the quick response. My apologies, that was a typo. I had a question about the ordering changing, but I think I figured it out so I omitted it and forgot to remove it. I also updated by recommendation.
> > > >
> > > > My question was about the relevance of similarity ordering on the implied structure. If you did explicit mergers, it would be based on passing a threshold as noted in the comments, and since values only increase, then mergers would never be split and the order of the similarity would be irrelevant to the implied structure. However, since you do soft-mergers, the ordering doesn't matter as well. Please correct if I am wrong.

---

> > > > > ### Author Response · Authors · 2022-11-18
> > > > > **Reply to the response**
> > > > >
> > > > > Dear Reviewer,
> > > > >
> > > > > Your understanding of the merge operation is right. Thanks for the discussion!

---

### Official Review · Reviewer_o933 · 2022-11-02

**Confidence:** 4
**Correctness:** 4
**Technical Novelty And Significance:** 3
**Empirical Novelty And Significance:** 4
**Recommendation:** 8

**Clarity, Quality, Novelty And Reproducibility:**

- Writing is clear, and the organization of manuscript is easy to follow, related work is well-surveyed.
- I think that CLIP models have shown significant progress recently, this work provides novel, plausible, concrete, and reasonable approaches and shows significant performance improvement as expected. I believe that researchers in the ICLR community would be interested in this work.
- The authors provide enough detail information to be reproducible.


**Strength And Weaknesses:**

*Strength
- The paper provides enough supports to convince the claims with diverse experimental settings of tasks, pretraining datasets, and downstream datasets.

*Weaknesses
- The authors present (in Appendices) some successful cases of the induced hierarchies. However, it is important to clarify the pros and cons of HiCLIP. I think that it would be more comprehensive to provide unintended results of HiCLIP.
- It is not clearly explained how the hierarchical features help to learn better representation. As one of suggestions, it might be helpful to comparatively visualize the feature spaces of CLIP, DeCLIP and HiCLIP.


**Summary Of The Paper:**

This paper proposes a novel specialized attention mechanism for CLIP models to induce hierarchical feature discovery in vision and language each modality.
The authors successfully validate their claims with various experimental results, and they demonstrate the significant performance improvement of CLIP (Contrastive vision-language pretraining) for 11 visual recognition tasks, image-text retrieval task and vision-language reasoning task such as VQAv2 and SNLI-VE.


**Summary Of The Review:**

See the above.

----------
**Update (Nov 26th)**

Since the authors resolve my major concerns, I've raised my score from 6 to 8.

---

> ### Author Response · Authors · 2022-11-17
> **For Reviewer o933**
>
> Dear reviewer:
>
> Thank you for the helpful comments.
>
> According to your suggestion, we comparatively visualize the feature spaces of CLIP, DeCLIP, and HiCLIP. As shown in **Figure 6 and Figure 7 in Section D of the Appendix**, we provide the t-SNE visualization of the learned feature space for CLIP, HiCLIP and DeCLIP pretrained on YFCC-15M and 30M data, respectively. We select the CIFAR-10 dataset to conduct the visualization experiments.
>
> From the figures, we can observe that HiCLIP achieves a more obvious separation of image features belonging to different categories than CLIP. For example, in both Figure 6 and Figure 7, while CLIP and DeCLIP cannot very well separate Category 8 (light green dots) and Category 0 (blue dots), HiCLIP achieves better separation. The results demonstrate that the adoption of hierarchical attention helps HiCLIP extract more distinguishable representations than CLIP. Following our intuition in Section 1, we argue that by explicitly modeling the hierarchical structure of images and texts, HiCLIP captures better visual and semantic embeddings that are better aligned, and hence achieves better performance in both vision tasks and vision-language tasks.
>
> We are still working on generating more visualization cases as well as a more detailed description of how the visualization results are generated. We will update them as well as clarify the pros and cons of HiCLIP soon.

---

> > ### Author Response · Authors · 2022-11-18
> > **For Reviewer o933 (Cont'd)**
> >
> > Dear reviewer:
> >
> > In our updated version PDF, we have provided six more comprehensive unsupervised induction cases of HiCLIP including both successful and unintended cases. In **Section C of the updated version of the paper PDF**, we also elaborate on how the visualization results are generated. We then clarify the pros and cons of HiCLIP as follows:
> >
> > Pros:
> > We integrate hierarchy-aware attention into the conventional attention mechanism, which can discover and aggregate spatially and semantically similar visual patches and language tokens in a layer-by-layer manner.
> > 1) With the help of hierarchy-aware attention, we can improve both CLIP and DeCLIP by a large margin on a variety of visual and vision-language downstream tasks.
> > 2) Based on the learned hierarchy-aware attention mask $C$ of all Transformer layers, we can induct tree and group hierarchies of input texts and images in an unsupervised style, respectively.
> > 3) According to the additional experiment, HiCLIP learns more distinguishable representations.
> >
> >
> > Cons:
> > The unsupervised hierarchy induction of HiCLIP (visualization of vision encoder especially) follows a top-down style and relies on the threshold values to decide whether to split two adjacent visual patches and language tokens. For the visual hierarchy, we trivially specify thresholds for different layers (the higher layer also has a higher threshold value). Thus, the threshold list may not be suitable for every image. In addition, changing the threshold values may influence the visual and language induction results. It would be better if the thresholds are adaptive to every input image and sentence. Our future work is to find a better way (e.g., a data-dependent algorithm) to parse the $C$ matrix for each layer.

---

> > > ### Comment · Reviewer_o933 · 2022-11-26
> > > **Reviewer Response**
> > >
> > > Thank you for the responses and the corresponding updating.
> > >
> > > The authors show qualitative results to clarify the issues of weak points.
> > > I think this revision makes the manuscript stronger and more persuasive, while it would be better to present quantitative scores.
> > >
> > > From the reason above, I've raised my score higher.
> > > I believe that researchers in the ICLR community would be interested in this work.

---

### Comment · Reviewer_WpY2 · 2022-11-15
**Initiating paper discussion**

Dear authors, reviewers, and AC,

I thought I would get some discussion started in this paper as I haven't seen any response from the authors and we're in the last portion of the discussion section. Overall, I liked the paper, but I had a few questions and concerns that I was hoping the authors (or other reviewers) could address.

My primary concern is with the clarity of the method section and less so, the missing comparisons. While I tentatively set my recommendation to borderline accept, I did so with the assumption that the authors could address my clarification requests. Without clarification (from authors or other reviewers), I think the current explanation of the method explanation seems unclear and I fail to see how some of the properties the authors stated are ensured by the method. If I missing something, I would appreciate it if the other reviewers could comment on this.

I second reviewers o933 and jjXQ questions regarding what the hierarchy is learning: what does it mean for the zebra and patch of grass/sky to be grouped together? are there any (expected/observed) limitations to what hierarchy can provide?  I think the coarse patching makes it unclear what patches belong together or not; eg, in Figure 3, should a patch that's half-plane/half-grass be grouped with the plane or the grass. Furthermore, for longer descriptions, one would expect noun phrases that include both an object and its context (eg, "a zebra grazing") resulting in valid patches that include zerba/patch. However, it's unclear if that was the case. It would be nice if the authors could provide some discussion of this.

---

> ### Author Response · Authors · 2022-11-18
> **Reply to the paper discussion**
>
> Dear reviewers and AC,
>
> We have provided responses to the questions and concerns of all reviewers. We have conducted additional experiments and added 1) comparisons with GroupViT, SLIP, FILIP, and DeFILIP; 2) 11 datasets average accuracy in Table 4; 3) unsupervised grammar induction results on MSCOCO.
>
> We also update our paper with more visualization results of unsupervised induction cases, visualization of the learned feature space (a comparison among CLIP, HiCLIP, and DeCLIP on CIFAR 10), a toy example to clarify how $C$ is computed, as well as the detailed description of the unsupervised hierarchy induction process. You can find all the above contents in the Appendix of our updated PDF.
>
> For unsupervised hierarchy induction of input images, the group operation in our HiCLIP considers both spatial and semantic similarities. The final output will be one $512$-dimension embedding representing the whole image. Therefore, the zebra/grass patches will eventually merge together.
>
> There is a Softmax operation in Eq. (7), where the patch at location $(i,j)$ needs to be compared with its four-adjacent neighboring patches and then normalized. In fact, it is possible that 1) patch $(i,j)$ are not similar to all its neighbors at all; 2) but say patch $(i,j)$ and patch $(i+1,j)$ are more similar. So after normalization, the similarity between patch $(i,j)$ and patch $(i+1,j)$ is higher.
>
> Our current hierarchy induction solution for the visual inputs is a natural 2D extension of what Tree Transformer does, which indicates that each patch will sooner or later be merged with its neighbors. In the future, we plan to test other possible solutions such as increasing the perception field of the Softmax operation (i.e., considering more patches when performing comparisons).

---

### Decision · Program_Chairs · 2023-01-20

**Decision:**

Accept: poster

**Justification For Why Not Higher Score:**

The paper is solid, but feels to me like an incremental improvement on CLIP.

**Justification For Why Not Lower Score:**

All reviewers felt the paper should be accepted.

**Metareview: Summary, Strengths And Weaknesses:**

The paper presents a new attention mechanism for CLIP-like methods which enables hierarchical feature learning. Initial reviews of the paper were mixed; reviewers wanted more discussion about the pros and cons of the method and the role of hierarchy in prior architectures; there were also issues with missing comparisons to prior methods (SLIP and FILIP), insufficient analysis of the Tree Transformer and Group Transformer, and many small details about the attention mechanism and experimental setup. The authors provided an in-depth rebuttal addressing these issues, and after the discussion phase all reviewers felt that their concerns had been largely addressed. In the end all reviewers felt that the paper should be accepted. The authors are strongly encouraged to take the feedback from all reviewers into account when preparing the camera-ready version of the paper.

**Note From Pc:**

if the above contains the word "oral" or "spotlight" please see: "oral" presentation means -> notable-top-5% and "spotlight" means -> notable-top-25%. As stated in our emails, we are disassociating presentation type from AC recommendations